# Online Convex Optimization with Unbounded Memory

**Raunak Kumar**
Department of Computer Science
Cornell University
Ithaca, NY 14853
raunak@cs.cornell.edu

**Sarah Dean**
Department of Computer Science
Cornell University
Ithaca, NY 14853
sdean@cornell.edu

**Robert Kleinberg**
Department of Computer Science
Cornell University
Ithaca, NY 14853
rdk@cs.cornell.edu

## Abstract

Online convex optimization (OCO) is a widely used framework in online learning. In each round, the learner chooses a decision in a convex set and an adversary chooses a convex loss function, and then the learner suffers the loss associated with their current decision. However, in many applications the learner's loss depends not only on the current decision but on the entire history of decisions until that point. The OCO framework and its existing generalizations do not capture this, and they can only be applied to many settings of interest after a long series of approximation arguments. They also leave open the question of whether the dependence on memory is tight because there are no non-trivial lower bounds. In this work we introduce a generalization of the OCO framework, "Online Convex Optimization with Unbounded Memory", that captures long-term dependence on past decisions. We introduce the notion of $p$-effective memory capacity, $H_p$, that quantifies the maximum influence of past decisions on present losses. We prove an $O(\sqrt{H_p T})$ upper bound on the policy regret and a matching (worst-case) lower bound. As a special case, we prove the first non-trivial lower bound for OCO with finite memory [Anava et al., 2015], which could be of independent interest, and also improve existing upper bounds. We demonstrate the broad applicability of our framework by using it to derive regret bounds, and to improve and simplify existing regret bound derivations, for a variety of online learning problems including online linear control and an online variant of performative prediction.

## 1 Introduction

Numerous applications are characterized by multiple rounds of sequential interactions with an environment, e.g., prediction from expert advice [Littlestone and Warmuth, 1989, 1994], portoflio selection [Cover, 1991], routing [Awerbuch and Kleinberg, 2008], etc. One of the most popular frameworks for modelling such sequential decision-making problems is online convex optimization (OCO) [Zinkevich, 2003]. The OCO framework is as follows. In each round, the learner chooses a decision in a convex set and an adversary chooses a convex loss function, and then the learner suffers the loss associated with their current decision. The performance of an algorithm is measured by regret: the difference between the algorithm's total loss and that of the best fixed decision. We refer the reader to Shalev-Shwartz [2012], Hazan [2019], Orabona [2019] for surveys on this topic.

37th Conference on Neural Information Processing Systems (NeurIPS 2023).

However, in many applications the loss of the learner depends not only on the current decisions but on the entire history of decisions until that point. For example, in online linear control [Agarwal et al., 2019b], in each round the learner chooses a "control policy" (i.e., decision), suffers a loss that is a function of the action taken by this policy and the current state of the system, and the system's state evolves according to linear dynamics. The current state depends on the entire history of actions and, therefore, the current loss depends not only on the current decision but the entire history of decisions. The OCO framework cannot capture such long-term dependence of the current loss on the past decisions and neither can existing generalizations that allow the loss to depend on a *constant* number of past decisions [Anava et al., 2015]. Although a series of approximation arguments can be used to apply finite memory generalizations of OCO to the online linear control problem, there is no OCO framework that captures the complete long-term dependence of current losses on past decisions. Furthermore, there are no non-trivial lower bounds for OCO in the memory setting,[1] which leaves open the question whether the dependence on memory is tight.

**Contributions.** In this paper we introduce a generalization of the OCO framework, "Online Convex Optimization with Unbounded Memory" (Section 2), that allows the loss in the current round to depend on the entire history of decisions until that point. We introduce the notion of $p$-effective memory capacity, $H_p$, that quantifies the maximum influence of past decisions on present losses. We prove an $O(\sqrt{H_p T})$ upper bound on the policy regret (Theorem 3.1) and a matching (worst-case) lower bound (Theorem 3.2). As a special case, we prove the first non-trivial lower bound for OCO with finite memory (Theorem 3.4), which could be of independent interest, and also improve existing upper bounds (Theorem 3.3). Our lower bound technique extends existing techniques developed for memoryless settings. We design novel adversarial loss functions that exploit the fact that an algorithm cannot overwrite its history. We illustrate the power of our framework by bringing together the regret analysis of two seemingly disparate problems under the same umbrella. First, we show how our framework improves and simplifies existing regret bounds for the online linear control problem [Agarwal et al., 2019b] in Theorem 4.1. Second, we show how our framework can be used to derive regret bounds for an online variant of performative prediction [Perdomo et al., 2020] in Theorem 4.2. This demonstrates the broad applicability of our framework for deriving regret bounds for a variety of online learning problems, particularly those that exhibit long-term dependence of current losses on past decisions.

**Related work.** The most closely related work to ours is the OCO with finite memory framework [Anava et al., 2015]. They consider a generalization of the OCO framework that allows the current loss to depend on a *constant* number of past decisions. There have been a number of follow-up works that extend the framework in a variety of other ways, such as non-stationarity [Zhao et al., 2022], incorporating switching costs [Shi et al., 2020], etc. However, none of these existing works go beyond a constant memory length and do not prove a non-trivial lower bound with a dependence on the memory length. In a different line of work, Bhatia and Sridharan [2020] consider a much more general online learning framework that goes beyond a constant memory length, but they only provide *non-constructive* upper bounds on regret. In contrast, our OCO with unbounded memory framework allows the current loss to depend on an *unbounded* number of past decisions, provides *constructive* upper bounds on regret, and lower bounds for a broad class of problems that includes OCO with finite memory with a general memory length $m$.

A different framework for sequential decision-making is multi-armed bandits [Bubeck and Cesa-Bianchi, 2012, Slivkins, 2019]. Qin et al. [2023] study a variant of contextual stochastic bandits where the current loss can depend on a sparse subset of all prior contexts. This setting differs from ours due to the feedback model, stochasticity, and decision space. Reinforcement learning [Sutton and Barto, 2018] is yet another popular framework for sequential decision-making that considers very general state-action models of feedback and dynamics. In reinforcement learning one typically measures regret with respect to the best state-action policy from some policy class, rather than the best fixed decision as in online learning and OCO. In the special case of linear control, policies can be reformulated as decisions while preserving convexity; we discuss this application in Section 4. Considering the general framework is an active area of research.

We defer discussion of related work for specific applications to Section 4.

---

[1]The trivial lower bound refers to the $\Omega(\sqrt{T})$ lower bound for OCO in the memoryless setting.

## 2 Framework

We begin with some motivation for the formalism used in our framework (Section 2.1). Many real-world applications involve controlling a physical dynamical system, for example, variable-speed wind turbines in wind energy electric power production [Boukhezzar and Siguerdidjane, 2010]. The typical solution for these problems has been to model them as offline control problems with linear time-invariant dynamics and use classical methods such as LQR and LQG [Boukhezzar and Siguerdidjane, 2010]. Instead of optimizing over the space of control inputs, the typical feedback control approach optimizes over the space of controllers, i.e., policies that choose a control input as a function of the system state. The standard controllers considered in the literature are linear controllers. Even when the losses are convex in the state and input, they are nonconvex in the linear controller. In the special case of quadratic losses in terms of the state and input, there is a closed-form solution for the optimal solution using the algebraic Riccati equations [Lancaster and Rodman, 1995]. But this does not hold for general convex losses resulting in convex reparameterizations such as Youla [Youla et al., 1976, Kučera, 1975] and SLS [Wang et al., 2019, Anderson et al., 2019]. The resulting parameterization represents an infinite dimensional system response and is characterized by a sequence of matrices. Recent work has studied an online approach for some of these control theory problems, where a sequence of controllers is chosen adaptively rather than choosing one offline [Abbasi-Yadkori and Szepesvári, 2011, Dean et al., 2018, Simchowitz and Foster, 2020, Agarwal et al., 2019b].

The takeaway from the above is that there are online learning problems in which (i) the current loss depends on the entire history of decisions; and (ii) the decision space can be more complicated than just a subset of $\mathbb{R}^d$, e.g., it can be an unbounded sequence of matrices. This motivates us to model the decision space as a Hilbert space and the history space as a Banach space in the formal problem setup below, and this subsumes the special cases of OCO and OCO with finite memory. This formalism not only lets us consider a wide range of spaces, such as $\mathbb{R}^d$, unbounded sequences of matrices, etc., but also lets us define appropriate norms on these spaces. This latter feature is crucial for deriving strong regret bounds for some applications such as online linear control. For this problem we derive improved regret bounds (Theorem 4.1) by defining weighted norms on the decision and history spaces, where the weights are chosen to leverage the problem structure.

**Notation.** We use $\|\cdot\|_{\mathcal{U}}$ to denote the norm associated with a space $\mathcal{U}$. The operator norm for a linear $L$ operator from space $\mathcal{U} \to \mathcal{V}$ is defined as $\|L\|_{\mathcal{U} \to \mathcal{V}} = \max_{u:\|u\|_{\mathcal{U}} \leq 1} \|Lu\|_{\mathcal{V}}$. For convenience, sometimes we simply use $\|\cdot\|$ when the meaning is clear from the context. For a finite-dimensional matrix we use $\|\cdot\|_F$ and $\|\cdot\|_2$ to denote its Frobenius norm and operator norm respectively.

### 2.1 Setup

Let the decision space $\mathcal{X}$ be a closed and convex subset of a Hilbert space $\mathcal{W}$ with norm $\|\cdot\|_{\mathcal{X}}$ and the history space $\mathcal{H}$ be a Banach space with norm $\|\cdot\|_{\mathcal{H}}$. Let $A : \mathcal{H} \to \mathcal{H}$ and $B : \mathcal{W} \to \mathcal{H}$ be linear operators. The game between the learner and an oblivious adversary proceeds as follows. Let $T$ denote the time horizon and $f_t : \mathcal{H} \to \mathbb{R}$ be loss functions chosen by the adversary. The initial history is $h_0 = 0$. In each round $t \in [T]$, the learner chooses $x_t \in \mathcal{X}$, the history is updated to $h_t = Ah_{t-1} + Bx_t$, and the learner suffers loss $f_t(h_t)$. An instance of an online convex optimization with unbounded memory problem is specified by the tuple $(\mathcal{X}, \mathcal{H}, A, B)$.

We use the notion of policy regret [Dekel et al., 2012] as the performance measure in our framework. The policy regret of a learner is the difference between its total loss and the total loss of a strategy that plays the best fixed decision in every round. The history after round $t$ for a strategy that chooses $x$ in every round is described by $h_t = \sum_{k=0}^{t-1} A^k B x$, which motivates the following definition.

**Definition 2.1.** Given $f_t : \mathcal{H} \to \mathbb{R}$, the function $\tilde{f}_t : \mathcal{X} \to \mathbb{R}$ is defined by $\tilde{f}_t(x) = f_t(\sum_{k=0}^{t-1} A^k B x)$.

**Definition 2.2** (Policy Regret). The policy regret of an algorithm $\mathcal{A}$ is defined as $R_T(\mathcal{A}) = \sum_{t=1}^{T} f_t(h_t) - \min_{x \in \mathcal{X}} \sum_{t=1}^{T} \tilde{f}_t(x)$.

In many motivating examples such as online linear control (Section 4.1), the history at the end of a round is a sequence of linear transformations of past decisions. The following definition captures this formally and we leverage this structure to prove stronger regret bounds (Theorem 3.1).

**Definition 2.3** (Linear Sequence Dynamics). Consider an online convex optimization with unbounded memory problem specified by $(\mathcal{X}, \mathcal{H}, A, B)$. Let $(\xi_k)_{k=0}^{\infty}$ be a sequence of nonnegative real numbers satisfying $\xi_0 = 1$. We say that $(\mathcal{X}, \mathcal{H}, A, B)$ follows linear sequence dynamics with the $\xi$-weighted $p$-norm for $p \geq 1$ if

1. $\mathcal{H}$ is the $\xi$-weighted $\ell^p$-direct sum of a finite or countably infinite number of copies of $\mathcal{W}$: every element $y \in \mathcal{H}$ is a sequence $y = (y_i)_{i \in \mathcal{I}}$, where $\mathcal{I} = \mathbb{N}$ or $\mathcal{I} = \{0, \ldots, n\}$ for some $n \in N$, and $\|y\|_{\mathcal{H}} = \left( \sum_{i \in \mathcal{I}} (\xi_i \|y_i\|)^p \right)^{1/p} < \infty$.

2. We have $A(y_0, y_1, \ldots) = (0, A_0 y_0, A_1 y_1, \ldots)$, where $A_i : \mathcal{W} \to \mathcal{W}$ are linear operators.

3. The operator $B$ satisfies $B(x) = (x, 0, \ldots)$.

Note that since the norm on $\mathcal{H}$ depends on the weights $\xi$, the operator norm $\|A^k\|$ also depends on $\xi$. If the weights are all equal to 1, then we simply say $p$-norm instead of $\xi$-weighted $p$-norm.

## 2.2 Assumptions

We make the following assumptions about the feedback model and the loss functions.

**A1** The learner knows the operators $A$ and $B$, and observes $f_t$ at the end of each round $t$.

**A2** The operator norm of $B$ is at most 1, i.e., $\|B\| \leq 1$.

**A3** The functions $f_t$ are convex.

**A4** The functions $f_t$ are $L$-Lipschitz continuous: $\forall\, h, \tilde{h} \in \mathcal{H}$ and $t \in [T]$, we have $|f_t(h) - f_t(\tilde{h})| \leq L \|h - \tilde{h}\|_{\mathcal{H}}$.

Regarding Assumption **A1**, our results easily extend to the case where instead of observing $f_t$, the learner receives a gradient $\nabla \tilde{f}_t(x_t)$ from a gradient oracle, which can be implemented using knowledge of $f_t$ and the dynamics $A$ and $B$. Handling the cases when the operators $A$ and $B$ are unknown and/or the learner observes bandit feedback (i.e., only $f_t(h_t)$) are important problems and we leave them as future work. Note that our assumption that $A$ and $B$ are known is no more restrictive than in the existing literature on OCO with finite memory [Anava et al., 2015] where it is assumed that the learner knows the constant memory length. In fact, our assumption is more general because our framework not only captures constant memory length as a special case but allows for richer dynamics as we illustrate in Section 4. Assumption **A2** is made for convenience, and it amounts to a rescaling of the problem. Assumption **A3** can be replaced by the *weaker* assumption that $\tilde{f}_t$ are convex (similar to the literature on OCO with finite memory [Anava et al., 2015]) and this is what we use in the rest of the paper.

Assumptions **A1** and **A4** imply that $\tilde{f}_t$ are $\tilde{L}$-Lipschitz continuous for the following $\tilde{L}$.

**Theorem 2.1.** *Consider an online convex optimization with unbounded memory problem specified by $(\mathcal{X}, \mathcal{H}, A, B)$. If $f_t$ is $L$-Lipschitz continuous, then $\tilde{f}_t$ is $\tilde{L}$-Lipschitz continuous for $\tilde{L} \leq L \sum_{k=0}^{\infty} \|A^k\|$. If $(\mathcal{X}, \mathcal{H}, A, B)$ follows linear sequence dynamics with the $\xi$-weighted $p$-norm for $p \geq 1$, then $\tilde{L} \leq L \left( \sum_{k=0}^{\infty} \|A^k\|^p \right)^{\frac{1}{p}}$.*

The proof follows from the definitions of $\tilde{f}_t$ and $\|\cdot\|_{\mathcal{H}}$, and we defer it to Appendix A. The above bound is tighter than similar results in the literature on OCO with finite memory and online linear control. This theorem is a key ingredient, amongst others, in improving existing upper bounds on regret for OCO with finite memory (Theorem 3.3) and for online linear control (Theorem 4.1). Before presenting our final assumption we introduce the notion of $p$-effective memory capacity that quantifies the maximum influence of past decisions on present losses.

**Definition 2.4** ($p$-Effective Memory Capacity). Consider an online convex optimization with unbounded memory problem specified by $(\mathcal{X}, \mathcal{H}, A, B)$. For $p \geq 1$, the $p$-effective memory capacity is defined as

$$H_p(\mathcal{X}, \mathcal{H}, A, B) = \left( \sum_{k=0}^{\infty} k^p \|A^k\|^p \right)^{\frac{1}{p}}. \tag{1}$$

When the meaning is clear from the context we simply use $H_p$ instead. The $p$-effective memory capacity is an upper bound on the difference in histories for two sequences of decisions whose difference grows at most linearly with time. To see this, consider two sequences of decisions, $(x_k)$ and $(\tilde{x}_k)$, whose elements differ by no more than $k$ at time $k$: $\|x_k - \tilde{x}_k\| \leq k$. Then the histories generated by the two sequences have difference between bounded as $\|h - \tilde{h}\| = \|\sum_k A^k B(x_k - \tilde{x}_k)\| \leq \sum_k k\|A^k B\| \leq \sum_k k\|A^k\| = H_1$, where the last inequality follows from Assumption **A2**. A similar bound holds with $H_p$ instead when $(\mathcal{X}, \mathcal{H}, A, B)$ follows linear sequence dynamics with the $\xi$-weighted $p$-norm.

    **A5** The 1-effective memory capacity is finite, i.e., $H_1 < \infty$.

Since $H_p$ is decreasing in $p$, $H_1 < \infty$ implies $H_p < \infty$ for all $p \geq 1$. For the case of linear sequence dynamics with the $\xi$-weighted $p$-norm it suffices to make the *weaker* assumption that $H_p < \infty$. However, for simplicity of exposition, we assume that $H_1 < \infty$.

### 2.3 Special Cases

**OCO with Finite Memory.** Consider the OCO with finite memory problem with constant memory length $m$. It can be specified in our framework by $(\mathcal{X}, \mathcal{H}, A_{\text{finite},m}, B_{\text{finite},m})$, where $\mathcal{H}$ is the $\ell^2$-direct sum of $m$ copies of $\mathcal{X}$, $A_{\text{finite},m}(x^{[m]}, \ldots, x^{[1]}) = (0, x^{[m]}, \ldots, x^{[2]})$, and $B_{\text{finite},m}(x) = (x, 0, \ldots, 0)$. Note that $(\mathcal{X}, \mathcal{H}, A_{\text{finite},m}, B_{\text{finite},m})$ follows linear sequence dynamics with the 2-norm. Our framework can even model an extension where the problem follows linear sequence dynamics with the $p$-norm for $p \geq 1$ by simply defining $\mathcal{H}$ to be the $\ell^p$-direct sum of $m$ copies of $\mathcal{X}$.

**OCO with $\rho$-discounted Infinite Memory.** Our framework can also model OCO with infinite memory problems that are not modelled by existing OCO frameworks. Let $\rho \in (0, 1)$ be the discount factor and $p \geq 1$. An OCO with $\rho$-discounted infinite memory problem is specified by $(\mathcal{X}, \mathcal{H}, A_{\text{infinite},\rho}, B_{\text{infinite},\rho})$, where $\mathcal{H}$ is the $\ell^p$-direct sum of countably many copies of $\mathcal{X}$, $A_{\text{infinite},\rho}((y_0, y_1, \ldots)) = (0, \rho y_0, \rho y_1, \ldots)$, and $B_{\text{infinite},\rho}(x) = (x, 0, \ldots)$. Note that $(\mathcal{X}, \mathcal{H}, A_{\text{infinite},\rho}, B_{\text{infinite},\rho})$ follows linear sequence dynamics with the $p$-norm. Due to space constraints we defer proofs of regret bounds for this problem to the appendix.

## 3 Regret Analysis

We present two algorithms for choosing the decisions $x_t$. Algorithm 1 uses follow-the-regularized-leader (FTRL) [Shalev-Shwartz and Singer, 2006, Abernethy et al., 2008] on the loss functions $\tilde{f}_t$. Due to space constraints, we discuss how to implement it efficiently in Appendix G and present simple simulation experiments in Appendix I. Algorithm 2, which we only present in Appendix H, combines FTRL with a mini-batching approach [Dekel et al., 2012, Altschuler and Talwar, 2018, Chen et al., 2020] to additionally guarantee that the decisions switch at most $O(T\tilde{L}/LH_1)$ times. We defer the proofs of the following upper and lower bounds to Appendices C and D respectively.

---

**Algorithm 1:** FTRL

**Input** : Time horizon $T$, step size $\eta$, $\alpha$-strongly-convex regularizer $R : \mathcal{X} \to \mathbb{R}$.
1 Initialize history $h_0 = 0$.
2 **for** $t = 1, 2, \ldots, T$ **do**
3      Learner chooses $x_t \in \arg\min_{x \in \mathcal{X}} \sum_{s=1}^{t-1} \tilde{f}_s(x) + \frac{R(x)}{\eta}$.
4      Set $h_t = A h_{t-1} + B x_t$.
5      Learner suffers loss $f_t(h_t)$ and observes $f_t$.
6 **end**

---

**Theorem 3.1.** *Consider an online convex optimization with unbounded memory problem specified by $(\mathcal{X}, \mathcal{H}, A, B)$. Let the regularizer $R : \mathcal{X} \to \mathbb{R}$ be $\alpha$-strongly-convex and satisfy $|R(x) - R(\tilde{x})| \leq D$ for all $x, \tilde{x} \in \mathcal{X}$. Algorithm 1 with step-size $\eta$ satisfies $R_T(\text{FTRL}) \leq \frac{D}{\eta} + \eta \frac{T\tilde{L}^2}{\alpha} + \eta \frac{TL\tilde{L}H_1}{\alpha}$. If*

$\eta = \sqrt{\frac{\alpha D}{T\tilde{L}(LH_1+\tilde{L})}}$, then

$$R_T(\textit{FTRL}) \leq O\left(\sqrt{\frac{D}{\alpha}TL\tilde{L}H_1}\right).$$

*When $(\mathcal{X}, \mathcal{H}, A, B)$ follows linear sequence dynamics with the $\xi$-weighted $p$-norm, then all of the above hold with $H_p$ instead of $H_1$.*

The proof of this theorem involves writing the regret as $\sum_t f_t(h_t) - \tilde{f}_t(x_t) + \sum_t \tilde{f}_t(x_t) - \tilde{f}_t(x^*)$, and bounding the first term using the $p$-effective memory capacity and the second term using FTRL. We defer the full proof to Appendix C. The following lower bound shows that this is tight in the worst-case.

**Theorem 3.2.** *There exists an instance of the online convex optimization with unbounded memory problem, $(\mathcal{X}, \mathcal{H}, A, B)$, that follows linear sequence dynamics with the $\xi$-weighted $p$-norm and there exist $L$-Lipschitz continuous loss functions $\{f_t : \mathcal{H} \to \mathbb{R}\}_{t=1}^T$ such that the regret of any algorithm $\mathcal{A}$ satisfies*

$$R_T(\mathcal{A}) \geq \Omega\left(\sqrt{TL\tilde{L}H_p}\right).$$

The proof of this theorem follows from our lower bound for the special case of OCO with finite memory (Theorem 3.4), which we discuss in detail below. However, as we show in Appendix D, the lower bound holds for a much broader class of problems.

**Specialization to OCO with finite memory.** Now we show how our bounds specialize to the special case of OCO with finite memory (Section 2.3). Due to space constraints, we defer the specialization of the upper and lower bounds for OCO with $\rho$-discounted infinite memory (Section 2.3) to Appendix C and Appendix D respectively.

**Theorem 3.3.** *Consider an online convex optimization with finite memory problem with constant memory length $m$ specified by $(\mathcal{X}, \mathcal{H} = \mathcal{X}^m, A_{finite,m}, B_{finite,m})$. Let the regularizer $R : \mathcal{X} \to \mathbb{R}$ be $\alpha$-strongly-convex and satisfy $|R(x) - R(\tilde{x})| \leq D$ for all $x, \tilde{x} \in \mathcal{X}$. Algorithm 1 with step-size $\eta = \sqrt{\frac{\alpha D}{T\tilde{L}(Lm^{\frac{3}{2}}+\tilde{L})}}$ satisfies*

$$R_T(\textit{FTRL}) \leq O\left(\sqrt{\frac{D}{\alpha}TL\tilde{L}m^{\frac{3}{2}}}\right) \leq O\left(m\sqrt{\frac{D}{\alpha}TL^2}\right).$$

This follows from using the definition of $A_{\text{finite},m}$ to bound $\tilde{L}$ and $H_2$ in Theorem 3.1. This improves existing results [Anava et al., 2015] by a factor of $m^{1/4}$. Our bound depends on the Lipschitz continuity constants as $\sqrt{L\tilde{L}}$ whereas existing bounds depend as $\tilde{L}$, and $\tilde{L}$ can be as large as $\sqrt{m}L$ (Theorem 2.1). We defer the full proof to Appendix C and a detailed comparison with existing results to Appendix C.1. The following lower bound shows that this is tight in the worst-case.

**Theorem 3.4.** *There exists an instance of the online convex optimization with finite memory problem with constant memory length $m$, $(\mathcal{X}, \mathcal{H} = \mathcal{X}^m, A_{finite,m}, B_{finite,m})$, and there exist $L$-Lipschitz continuous loss functions $\{f_t : \mathcal{H} \to \mathbb{R}\}_{t=1}^T$ such that the regret of any algorithm $\mathcal{A}$ satisfies*

$$R_T(\mathcal{A}) \geq \Omega\left(m\sqrt{TL^2}\right).$$

To the best of our knowledge, this is the first non-trivial lower bound for OCO with finite memory with an explicit dependence on the memory length $m$. Our construction involves three main steps, the first two of which are loosely based on Altschuler and Talwar [2018]. First, divide time into $N = T/m$ blocks of size $m$. (For simplicity, assume $T$ is a multiple of $m$.) Second, sample a random sign $\epsilon_n$ for each block $n \in [N]$. Third, for $t > m$ choose

$$f_t(h_t) = \underbrace{\epsilon_{\lceil \frac{t}{m} \rceil}}_{(a)} \underbrace{Lm^{-\frac{1}{2}}}_{(b)} \underbrace{\left(x_{t-m+1} + \cdots + x_{m\lfloor \frac{t}{m} \rfloor + 1}\right)}_{(c)},$$

where term (a) is the random sign $\epsilon_n$ sampled for the block $n = \lceil t/m \rceil$ that $t$ belongs to, term (b) is a scaling factor chosen while respecting the Lipschitz continuity constraint, and term (c) is a sum over a *subset* of past decisions. Two important features of this construction are: (i) a random sign is sampled for each block rather than each round; and (ii) the loss in round $t$ depends on the history of decisions until and including the first round of the block that $t$ belongs to. These exploit the fact that an algorithm cannot overwrite its history and penalize it for its past decisions even after it observes the random sign $\epsilon_n$ for the current block. (See Fig. 1 for an illustration.) Existing lower bound proofs for OCO sample a random sign in each round and choose $f_t(x_t) \propto \epsilon_t x_t$. A first attempt at extending this for the OCO with finite memory setting would be to choose $f_t(h_t) \propto \epsilon_t \sum_{k=0}^{m-1} x_{t-k}$. However, in constrast to our approach, this does not exploit the fact that an algorithm cannot overwrite its history and does not suffice for obtaining a matching lower bound.

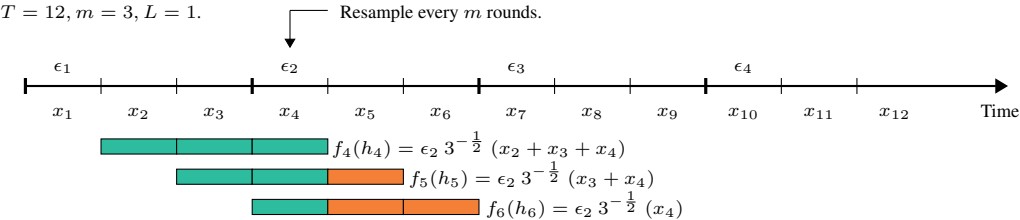

Figure 1: An illustration of the loss functions $f_t$ for the OCO with finite memory lower bound.

**Comparison of upper bound with prior work.** The algorithmic ideas and analysis for our regret upper bound are influenced by Anava et al. [2015]. However, an important innovation in our work is the use of weighted norms in the case of linear sequence dynamics. This is a simple but powerful way of encoding prior knowledge about a problem, and allows us to derive non-trivial regret bounds in the case of unbounded-length histories. The technical complications that arise are captured in bounding the relevant quantities of interest, e.g., the Lipschitz constant $\tilde{L}$, the operator norm $\|A^k\|$, etc. Furthermore, using weighted norms even leads to improved regret bounds for some applications. Indeed, consider the application to online linear control with adversarial disturbances (Section 4.1). Our framework and upper bound applied to this problem (Theorem 4.1) improve upon the existing upper bound, which used a finite memory approximation. See Lemmas E.2 and E.6 for an illustration of the technical details involved when using weighted norms.

## 4 Applications

In this section we apply our framework to online linear control (Section 4.1) and online performative prediction (Section 4.2). We defer expanded details and proofs to Appendices E and F respectively.

### 4.1 Online Linear Control

**Background.** Online linear control (OLC) is the problem of controlling a system with linear dynamics, adversarial disturbances, and adversarial and convex losses. It combines aspects from control theory and online learning. We refer the reader to Agarwal et al. [2019b] for more details. Here, we introduce the basic mathematical setup of the problem.

Let $\mathcal{S} \subseteq \mathbb{R}^{d_s}$ and $\mathcal{U} \subseteq \mathbb{R}^{d_u}$ denote the state and control spaces. Let $s_t$ and $u_t$ denote the state and control at time $t$ with $s_0$ being the inital state. The system evolves according to the linear dynamics $s_{t+1} = F s_t + G u_t + w_t$, where $F \in \mathbb{R}^{d_s \times d_s}, G \in \mathbb{R}^{d_s \times d_u}$ are matrices satisfying $\|F\|_2, \|G\|_2 \leq \kappa$ and $w_t \in \mathbb{R}^{d_s}$ is an adversarially chosen disturbance with $\|w\|_2 \leq W$. Without loss of generality, we assume that $\kappa, W \geq 1$, $d_s = d_u = d$, and also define $w_{-1} = s_0$. For $t = 0, \ldots, T-1$, let $c_t : \mathcal{S} \times \mathcal{U} \to [0, 1]$ be convex loss functions chosen by an oblivous adversary. The functions $c_t$ satisfy the following Lipschitz condition: if $\|s\|_2, \|u\|_2 \leq D_{\mathcal{X}}$, then $\|\nabla_s c_t(s, u)\|, \|\nabla_u c_t(s, u)\| \leq L_0 D_{\mathcal{X}}$. The goal in online linear control is to choose a sequence of policies that yield a sequence of controls $u_t$ to minimize the regret $R_T(\Pi) = \sum_{t=0}^{T-1} c_t(s_t, u_t) - \min_{\pi^* \in \Pi} \sum_{t=0}^{T-1} c_t(s_t^{\pi^*}, u_t^{\pi^*})$, where $s_t$ evolves according to linear dynamics stated above, $\Pi$ denotes a controller class, and $s_t^{\pi^*}, c_t^{\pi^*}$ denote the state and control at time $t$ when the controls are chosen according to $\pi^*$.

A very simple controller class is constant input, i.e., $\Pi = \{\pi_u : \pi(s) = u \in \mathcal{U}\}$. In this case, the history $h_t$ can be represented by the finite-dimensional state $s_t$, and the operators can be set to $A = F$ and $B = {}^G/_{\|G\|}$. However, like previous work [Agarwal et al., 2019b] we focus on the class of $(\kappa, \rho)$-strongly stable linear controllers, $\mathcal{K}$, where $K \in \mathcal{K}$ satisfies $F - GK = HLH^{-1}$ with $\|K\|_2, \|H\|_2, \|H^{-1}\|_2 \le \kappa$ and $\|L\|_2 = \rho < 1$. Given such a controller, the inputs are chosen as linear functions of the current state, i.e., $u_t = -Ks_t$. Unfortunately, parameterizing $u_t$ directly with a linear controller as $u_t = -Ks_t$ leads to a non-convex problem because $s_t$ is a non-linear function of $K$, e.g., if disturbances are 0, then $s_t = (F - GK)^t s_0$. An alternative parameterization is the disturbance-action controller (DAC).

**Definition 4.1.** Let $K \in \mathcal{K}$ be fixed. The class of disturbance-action controllers (DACs) $\mathcal{M}_K$ is $\{(K, M) : M = (M^{[s]})_{s=0}^{\infty}\}$, where $M^{[s]} \in \mathbb{R}^{d \times d}$ satisfies $\|M^{[s]}\|_2 \le \kappa^4 \rho^s$. The control in round $k$ is chosen as $u_k = -Ks_k + \sum_{s=1}^{k+1} M^{[s]} w_{k-s}$.

The class of such DACs has two important properties. First, it acts on the entire history of past disturbances. Consequently, given an arbitrary $K \in \mathcal{K}$, every $K^* \in \mathcal{K}$ can be expressed as a DAC $(K, M) \in \mathcal{M}_K$ with $M = (M^{[1]}, \dots, M^{[T+1]}, 0, \dots)$ [Agarwal et al., 2019a, Section 16.5]. That is, $\mathcal{K} \subseteq \mathcal{M}_K$ and it suffices to compute regret against $\mathcal{M}_K$ instead of $\mathcal{K}$. For the rest of this paper we fix $K \in \mathcal{K}$ and denote $\widetilde{F} = F - GK$. Second, suppose $M_t = (M_t^{[s]})_{s=0}^{\infty}$ is the parameter chosen in round $t$ and the control $u_t$ is chosen according to the DAC $(K, M_t)$. Then, $s_t$ and $u_t$ are *linear* functions of the parameters, which implies that $c_t$ is convex in the parameters. (See the next paragraph on "Formulation as OCO with Unbounded Memory" for a formula.) A similar parameterization was first considered for online linear control by Agarwal et al. [2019b] and is based on similar ideas in control theory, e.g., Youla [Youla et al., 1976, Kučera, 1975] and SLS [Wang et al., 2019, Anderson et al., 2019].

**Formulation as OCO with Unbounded Memory.**  The first step is a change of variables with respect to the control inputs from linear controllers to DACs and the second is a corresponding change of variables for the state. Define the decision space $\mathcal{X} = \{M = (M^{[s]}) : M^{[s]} \in \mathbb{R}^{d \times d}, \|M^{[s]}\|_2 \le \kappa^4 \rho^s\}$. Define the history space $\mathcal{H}$ to be the set consisting of sequences $h = (Y_k)$, where $Y_0 \in \mathcal{X}$ and $Y_k = \widetilde{F}^{k-1} G X_k$ for $X_k \in \mathcal{X}, k \ge 1$. (Recall $\widetilde{F} = F - GK$.) Define weighted norms $\|M\|_{\mathcal{X}}^2 = \sum_{s=1}^{\infty} \rho^{-s} \|M^{[s]}\|_F^2$ and $\|h\|_{\mathcal{H}}^2 = \sum_{k=0}^{\infty} \xi_k^2 \|Y_k\|_{\mathcal{X}}^2$, where the weights $(\xi_k)$ are defined as $\xi = (1, 1, 1, \rho^{-\frac{1}{2}}, \rho^{-1}, \rho^{-\frac{3}{2}}, \dots)$. Define the linear operators $A : \mathcal{H} \to \mathcal{H}$ and $B : \mathcal{W} \to \mathcal{H}$ as $A((Y_0, Y_1, \dots)) = (0, GY_0, \widetilde{F}Y_1, \widetilde{F}Y_2, \dots)$ and $B(M) = (M, 0, 0, \dots)$. Note that the problem follows linear sequence dynamics with the $\xi$-weighted 2-norm (Definition 2.3). The weights in the norms on $\mathcal{X}$ and $\mathcal{H}$ increase exponentially. However, the norms $\|M^{[s]}\|_F^2$ and $\|\widetilde{F}^{k-1}G\|_F^2$ decrease exponentially as well: by definition of $\mathcal{X}$ and the assumption on $\widetilde{F} = F - GK$ for $K \in \mathcal{K}$. Leveraging these exponential decays to define exponentially increasing weights is crucial for deriving our regret bounds that are stronger than existing results.

Construct the functions $f_t : \mathcal{H} \to \mathbb{R}$ that correspond to $c_t(s_t, u_t)$ as follows. Given a sequence of decisions $(M_0, \dots, M_t)$, the history at the end of round $t$ is given by $h_t = (M_t, GM_{t-1}, \widetilde{F}GM_{t-2}, \dots, \widetilde{F}^{t-1}GM_0, 0, \dots)$. A simple inductive argument shows that the state and control in round $t$ can be written as functions of $h_t$ as $s_t = \widetilde{F}^t s_0 + \sum_{k=0}^{t-1} \sum_{s=1}^{k+1} \widetilde{F}^{t-k-1} GM_k^{[s]} w_{k-s} + w_{t-1}$ and $u_t = -Ks_t + \sum_{s=1}^{t+1} M_t^{[s]} w_{t-s}$. Define the functions $f_t : \mathcal{H} \to \mathbb{R}$ by $f_t(h) = c_t(s, u)$, where $s$ and $u$ are the state and control determined by the history as above. Note that $f_t$ is parameterized by the past disturbances. Since the state and control are linear functions of the history and $c_t$ is convex, this implies that $f_t$ is convex. Now, given the above formulation and the fact that the class of disturbance-action controllers is a superset of the class of $(\kappa, \rho)$-strongly-stable linear controllers, we have that the policy regret for the online linear control problem is at most the policy regret, $\sum_{t=0}^{T-1} f_t(h_t) - \min_{M \in \mathcal{X}} \sum_{t=0}^{T-1} \tilde{f}_t(M)$. The following is our main result for online linear control and it improves existing results [Agarwal et al., 2019b] by a factor of $O(d \log(T)^{3.5} \kappa^5 (1 - \rho)^{-1})$. See Appendix E.3 for a detailed comparison.

**Theorem 4.1.** *Consider the online linear control problem as defined in Section 4.1. Suppose the decisions in round $t$ are chosen using Algorithm 1. Then, the upper bound on the policy regret is*

$$O\left(L_0 W^2 \sqrt{T} d^{\frac{1}{2}} \kappa^{17} (1 - \rho)^{-4.5}\right). \tag{2}$$

**Comparison with prior and concurrent work.** Existing works solve OLC (and its extensions) by making multiple finite memory approximations. First, they formulate the problem as OCO with finite memory. This requires bounding numerous error terms because the problem is inherently an OCO with unbounded memory problem. We bypass these error analysis steps entirely because the problem fits into our framework naturally. Second, existing works use the parameterization from Agarwal et al. [2019b] that only acts on a fixed, constant number of past disturbances. In particular, existing works use a "truncated" DAC policy that is a sequence of $d \times d$ matrices of length $2\kappa^4(1 - \rho)^{-1} \log T$. Our DAC policy acts on the entire history of disturbances and is a sequence of $d \times d$ matrices of unbounded length. Yet, we capture the dimension of this infinite-dimensional space in a way that still improves the overall bound, including completely eliminating the dependence on $\log T$, and improving the dependence on $d, \kappa$, and $(1 - \rho)$. This improvement comes from our novel use of weighted norms on the history and decision spaces. These norms allow us to give tighter bounds on the relevant quantities in the regret upper bound, e.g., $\|A^k\|$ (Lemma E.2) and $\tilde{L}$ (Lemma E.6).

In complementary concurrent work, Lin et al. [2022] focus on a more general online control problem. They improve regret bounds for this general version by a factor of $\log T$ compared to existing reductions to OCO with finite memory. They do so by using that the impact of a past policy decays geometrically with time. On the other hand, the primary focus of our work is studying the complete dependence of present losses on the entire history in OCO. Applying our resulting OCO with unbounded memory framework to OLC, we improve upon existing results for OLC by removing all $\log T$ factors and improving the dependence on $d, \kappa$, and $(1 - \rho)$.

## 4.2 Online Performative Prediction

**Background.** In many applications of machine learning the algorithm's decisions influence the data distribution, e.g., online labor markets [Anagnostopoulos et al., 2018, Horton, 2010], predictive policing [Lum and Isaac, 2016], on-street parking [Dowling et al., 2020, Pierce and Shoup, 2018], vehicle sharing markets [Banerjee et al., 2015], etc. Motivated by such applications, several works have studied the problem of performative prediction, which models the data distribution as a function of the decision-maker's decision [Perdomo et al., 2020, Mendler-Dünner et al., 2020, Miller et al., 2021, Brown et al., 2022, Ray et al., 2022, Jagadeesan et al., 2022]. Most of these works view the problem as a stochastic optimization problem; Jagadeesan et al. [2022] adopt a regret minimization perspective. We refer the reader to these citations for more details. As a natural extension to existing works, we introduce an online learning variant of performative prediction with geometric decay [Ray et al., 2022] that differs from the original formulations in a few key ways.

Let the decision set $\mathcal{X} \subseteq \mathbb{R}^d$ be closed and convex with $\|x\|_2 \leq D_{\mathcal{X}}$. Let $p_1$ denote the initial data distribution over the instance space $\mathcal{Z}$. In each round $t \in [T]$, the learner chooses a decision $x_t \in \mathcal{X}$ and an oblivious adversary chooses a loss function $l_t : \mathcal{X} \times \mathcal{Z} \to [0, 1]$, and then the learner suffers the loss $L_t(x_t) = \mathbb{E}_{z \sim p_t} [l_t(x_t, z)]$, where $p_t = p_t(x_1, \ldots, x_t)$ is the data distribution in round $t$. The goal in our online learning setting is to minimize the difference between the algorithm's total loss and the total loss of the best fixed decision, $\sum_{t=1}^{T} \mathbb{E}_{z \sim p_t} [l_t(x_t, z)] - \min_{x \in \mathcal{X}} \sum_{t=1}^{T} \mathbb{E}_{z \sim p_t(x)} [l_t(x, z)]$, where $p_t(x) = p_t(x, \ldots, x)$ is the data distribution in round $t$ had $x$ been chosen in all rounds so far. This measure is similar to performative regret [Jagadeesan et al., 2022] and is a natural generalization of performative optimality [Perdomo et al., 2020] for an online learning formulation.

We make the following assumptions. First, the loss functions $l_t$ are convex and $L_0$-Lipschitz continuous. Second, the data distribution satisfies for all $t \geq 1$, $p_{t+1} = \rho p_t + (1 - \rho)\mathcal{D}(x_t)$, where $\rho \in (0, 1)$ and $\mathcal{D}(x_t)$ is a distribution over $\mathcal{Z}$ that depends on the decision $x_t$ [Ray et al., 2022]. Third, $\mathcal{D}(x)$ is a location-scale distribution: $z \sim \mathcal{D}(x)$ iff $z \sim \xi + Fx$, where $F \in \mathbb{R}^{d \times d}$ satisfies $\|F\|_2 < \infty$ and $\xi$ is a random variable with mean $\mu$ and covariance $\Sigma$ [Ray et al., 2022].

Our problem formulation differs from existing work in the following ways. First, we adopt an online learning perspective on performative prediction with geometric decay, whereas Ray et al. [2022] adopt a stochastic optimization one. So, we assume that the loss functions $l_t$ are adversarially chosen, whereas Ray et al. [2022] assume $l_t = l$ are fixed. Second, we assume that the dynamics ($\mathcal{D}$ and $\rho$) are known (Assumption **A1**), whereas Ray et al. [2022] assume they are unknown and use samples from the data distribution. We believe that an appropriate extension of our framework that can deal with unknown linear operators $A$ and $B$ can be applied to this more difficult setting, and we leave this as future work. Third, even though Jagadeesan et al. [2022] also study an online learning variant

of performative prediction, they assume $l_t = l$ are fixed and the data distribution depends only on the current decisions, whereas we assume the data distribution depends on the entire history of decisions.

**Formulation as OCO with Unbounded Memory.** Let $\rho \in (0, 1)$. Let the decision space $\mathcal{X} \subseteq \mathbb{R}^d$ be closed and convex with the 2-norm. Let the history space $\mathcal{H}$ be the $\ell^1$-direct sum of countably infinite number of copies of $\mathcal{X}$. Define the linear operators $A : \mathcal{H} \to \mathcal{H}$ and $B : \mathcal{W} \to \mathcal{H}$ as $A((y_0, y_1, \dots)) = (0, \rho y_0, \rho y_1, \dots)$ and $B(x) = (x, 0, \dots)$. The problem is an OCO with $\rho$-discounted infinite memory problem and follows linear sequence dynamics with the 1-norm (Definition 2.3). Given a sequence of decisions $(x_k)_{k=1}^t$, the history is $h_t = (x_t, \rho x_{t-1}, \dots, \rho^{t-1} x_1, 0, \dots)$ and the data distribution $p_t = p_t(h_t)$ satisfies: $z \sim p_t$ iff $z \sim \sum_{k=1}^{t-1}(1-\rho)\rho^{k-1}(\xi + Fx_{t-k}) + \rho^t p_1$. This follows from the recursive definition of $p_t$ and parametric assumption about $\mathcal{D}(x)$. Define the functions $f_t : \mathcal{H} \to [0, 1]$ by $f_t(h_t) = \mathbb{E}_{z \sim p_t}[l_t(x_t, z)]$. Now, the original goal of minimizing the difference between the algorithm's total loss and the total loss of the best fixed decision is equivalent to minimizing the policy regret. The following is our main result for online performative prediction.

**Theorem 4.2.** *Consider the online performative prediction problem as defined in Section 4.2. Suppose the decisions in round $t$ are chosen using Algorithm 1. Then, the upper bound on the policy regret is*

$$O\left(D_{\mathcal{X}} L_0 \sqrt{T} \|F\|_2 (1-\rho)^{-\frac{1}{2}} \rho^{-1}\right).$$

# 5  Conclusion

In this paper we introduced a generalization of the OCO framework, "Online Convex Optimization with Unbounded Memory", that allows the loss in the current round to depend on the entire history of decisions until that point. We proved matching upper and lower bounds on the policy regret in terms of the time horizon, the $p$-effective memory capacity (a quantitative measure of the influence of past decisions on present losses), and other problem parameters (Theorems 3.1 and 3.2). As a special case, we proved the first non-trivial lower bound for OCO with finite memory (Theorem 3.4), which could be of independent interest, and also improved existing upper bounds (Theorem 3.3). We illustrated the power of our framework by bringing together the regret analysis of two seemingly disparate problems under the same umbrella: online linear control (Theorem 4.1), where we improve and simplify existing regret bounds, and online performative prediction (Theorem 4.2).

There are a number of directions for future research. A natural follow-up is to consider unknown dynamics (i.e., when the learner does not know the operators $A$ and $B$) and/or the case of bandit feedback (i.e., when the learner only observes $f_t(h_t)$). The extension to bandit feedback has been considered in the OCO and OCO with finite memory literature [Hazan and Li, 2016, Bubeck et al., 2021, Zhao et al., 2021, Gradu et al., 2020, Cassel and Koren, 2020]. It is tempting to think about a version where the history is a *nonlinear*, but decaying, function of the past decisions. The obvious challenge is that the nonlinearity would lead to non-convex losses. It is unclear how to deal with such issues, e.g., restricted classes of nonlinearities for which the OCO with unbounded memory perspective is still relevant [Zhang et al., 2015], different problem formulations such as online non-convex learning [Gao et al., 2018, Suggala and Netrapalli, 2020], etc.

There is a growing body of work on online linear control and its variants that rely on OCO with finite memory [Hazan et al., 2020, Agarwal et al., 2019c, Foster and Simchowitz, 2020, Cassel and Koren, 2020, Gradu et al., 2020, Li et al., 2021, Minasyan et al., 2021]. In this paper we showed how our framework can be used to improve and simplify regret bounds for the online linear control problem. Another direction for future work is to use our framework, perhaps with suitable extensions outlined above, to derive similar improvements for these other variants of online linear control.

## Acknowledgments and Disclosure of Funding

We thank Sloan Nietert and Victor Sanches Portella for helpful discussions. We thank Wei-Yu Chan for pointing out an error in the proof of Lemma C.1. We also thank anonymous AISTATS and NeurIPS reviewers for their helpful comments on an older and the current version of the paper respectively. This research was partially supported by the NSERC Postgraduate Scholarships-Doctoral Fellowship 545847-2020, NSF awards CCF-2312774 and OAC-2311521, and a gift from Wayfair.

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

## Organization of the Appendix

The appendix is organized as follows:

- Appendix A contains the proofs of the results from Section 2.
- Appendix B contains the proofs of existing results about FTRL for completeness.
- Appendix C contains the proofs of upper bounds on regret from Section 3.
- Appendix D contains the proofs of lower bounds on regret from Section 3.
- Appendix E contains the proofs of results about online linear control from Section 4.1.
- Appendix F contains the proofs of results about online performative prediction from Section 4.2.
- Appendix G discusses how to implement Algorithm 1 efficiently.
- Appendix H presents an algorithm for OCO with unbounded memory that provides the same upper bound on policy regret as Algorithm 1 while guaranteeting a small number of switches (Algorithm 2).
- Appendix I presents simulation experiments.

## A  Framework

In this section prove Theorem 2.1. But first we prove a lemma that we use for proofs involving linear sequence dynamics with the $\xi$-weighted $p$-norm (Definition 2.3). Recall that $\|\cdot\|_{\mathcal{U}}$ denotes the norm associated with a space $\mathcal{U}$ and the operator norm $\|L\|$ for a linear operator $L : \mathcal{U} \to \mathcal{V}$ is defined as $\|L\| = \max_{u:\|u\|_{\mathcal{U}} \leq 1} \|Lu\|_{\mathcal{V}}$.

**Lemma A.1.** *Consider an online convex optimization with unbounded memory problem specified by $(\mathcal{X}, \mathcal{H}, A, B)$. If $(\mathcal{X}, \mathcal{H}, A, B)$ follows linear sequence dynamics with the $\xi$-weighted $p$-norm for $p \geq 1$, then for all $k \geq 1$*

$$\xi_k \|A_{k-1} \cdots A_0\| \leq \|A^k\|.$$

*Proof.* Let $x \in \mathcal{X}$ with $\|x\|_{\mathcal{X}} = 1$. We have

$$\xi_k \|A_{k-1} \cdots A_0 x\|_{\mathcal{X}} = \|A^k(x, 0, \dots)\|_{\mathcal{H}} \leq \|A^k\| \|(x, 0, \dots)\|_{\mathcal{H}} \leq \|A^k\|,$$

where the last inequality follows because $\|(x, 0, \dots)\|_{\mathcal{H}} = \xi_0 \|x\|_{\mathcal{X}}$ and $\xi_0 = 1$ by Definition 2.3. Therefore, $\|A_{k-1} \cdots A_0\| \leq \|A^k\|$. ∎

**Theorem 2.1.** *Consider an online convex optimization with unbounded memory problem specified by $(\mathcal{X}, \mathcal{H}, A, B)$. If $f_t$ is $L$-Lipschitz continuous, then $\tilde{f}_t$ is $\tilde{L}$-Lipschitz continuous for $\tilde{L} \leq L \sum_{k=0}^{\infty} \|A^k\|$. If $(\mathcal{X}, \mathcal{H}, A, B)$ follows linear sequence dynamics with the $\xi$-weighted $p$-norm for $p \geq 1$, then $\tilde{L} \leq L \left( \sum_{k=0}^{\infty} \|A^k\|^p \right)^{\frac{1}{p}}$.*

*Proof.* Let $x, \tilde{x} \in \mathcal{X}$. For the general case, we have

$$
\left| \tilde{f}_t(x) - \tilde{f}_t(\tilde{x}) \right| = \left| f_t \left( \sum_{k=0}^{t-1} A^k B x \right) - f_t \left( \sum_{k=0}^{t-1} A^k B \tilde{x} \right) \right| \qquad \text{by Definition 2.1}
$$

$$
\leq L \left\| \sum_{k=0}^{t-1} A^k B (x - \tilde{x}) \right\|_{\mathcal{H}} \qquad f_t \text{ is } L\text{-Lipschitz continuous}
$$

$$
\leq L \sum_{k=0}^{t-1} \|A^k\| \|B\| \|x - \tilde{x}\|_{\mathcal{X}}
$$

$$
\leq L \sum_{k=0}^{t-1} \|A^k\| \|x - \tilde{x}\|_{\mathcal{X}} \qquad \text{by Assumption A2}
$$

$$
\leq L \sum_{k=0}^{\infty} \|A^k\| \|x - \tilde{x}\|_{\mathcal{X}}.
$$

If $(\mathcal{H}, \mathcal{X}, A, B)$ follows linear sequence dynamics with the $\xi$-weighted $p$-norm for $p \geq 1$, then we have

$$
\begin{aligned}
\left| \tilde{f}_t(x) - \tilde{f}_t(\tilde{x}) \right| &= \left| f_t \left( \sum_{k=0}^{t-1} A^k B x \right) - f_t \left( \sum_{k=0}^{t-1} A^k B \tilde{x} \right) \right| && \text{by Definition 2.1} \\
&\leq L \left\| \sum_{k=0}^{t-1} A^k B (x - \tilde{x}) \right\|_{\mathcal{H}} && f_t \text{ is } L\text{-Lipschitz continuous} \\
&= L \, \| (0, A_0(x - \tilde{x}), A_1 A_0(x - \tilde{x}), \dots) \| && \text{by Definition 2.3} \\
&= L \left( \sum_{k=0}^{t-1} \xi_k^p \, \| A_{k-1} \cdots A_0(x - \tilde{x}) \|^p \right)^{\frac{1}{p}} && \text{by Definition 2.3} \\
&\leq L \left( \sum_{k=0}^{t-1} \| A^k \|^p \right)^{\frac{1}{p}} \| x - \tilde{x} \|_{\mathcal{X}} && \text{by Lemma A.1} \\
&\leq L \left( \sum_{k=0}^{\infty} \| A^k \|^p \right)^{\frac{1}{p}} \| x - \tilde{x} \|_{\mathcal{X}} . \qquad \blacksquare
\end{aligned}
$$

## B  Standard Analysis of Follow-the-Regularized-Leader

In this section we state and prove some existing results about the follow-the-regularized-leader (FTRL) algorithm [Shalev-Shwartz and Singer, 2006, Abernethy et al., 2008]. These results are well known in the literature, but we prove them here for completeness and use them in the remainder of the paper. We use the below results for functions $\tilde{f}_t$ with Lipschitz constants $\tilde{L}$. However, in this section we use a more general notation, denoting functions by $g_t$ and their Lipschitz constant by $L_g$.

Consider the following setup for an online convex optimization (OCO) problem. Let $T$ denote the time horizon. Let the decision space $\mathcal{X}$ be a closed, convex subset of a Hilbert space and $g_t : \mathcal{X} \to \mathbb{R}$ be loss functions chosen by an oblivious adversary. The functions $g_t$ are convex and $L_g$-Lipschitz continuous. The game between the learner and the adversary proceeds as follows. In each round $t \in [T]$, the learner chooses $x_t \in \mathcal{X}$ and the learner suffers loss $g_t(x_t)$. The goal of the learner is to minimize (static) regret,

$$
R_T^{\text{static}} = \sum_{t=1}^{T} g_t(x_t) - \min_{x \in \mathcal{X}} \sum_{t=1}^{T} g_t(x). \tag{3}
$$

Let $R : \mathcal{X} \to \mathbb{R}$ be an $\alpha$-strongly convex regularizer satisfying $|R(x) - R(\tilde{x})| \leq D$ for all $x, \tilde{x} \in \mathcal{X}$. The FTRL algorithm chooses iterates $x_t$ as

$$
x_t \in \arg\min_{x \in \mathcal{X}} \sum_{s=1}^{t-1} g_s(x) + \frac{R(x)}{\eta}, \tag{4}
$$

where $\eta$ is a tunable parameter referred to as the step-size. In what follows, let $g_0 = \frac{R}{\eta}$. The analysis in this section closely follows Karlin [2017].

**Lemma B.1.** *For all $x \in \mathcal{X}$, FTRL (Eq. (4)) satisfies*

$$
\sum_{t=0}^{T} g_t(x) \geq \sum_{t=0}^{T} g_t(x_{t+1}).
$$

*Proof.* We use proof by induction on $T$. The base case is $T = 0$. By definition, $x_1 \in \arg\min_{x \in \mathcal{X}} R(x)$. Therefore, $R(x) \geq R(x_1)$ for all $x \in \mathcal{X}$. Recalling the notation $g_0 = \frac{R}{\eta}$ proves the base case. Now, assume that the lemma is true for $T - 1$. That is,

$$
\sum_{t=0}^{T-1} g_t(x) \geq \sum_{t=0}^{T-1} g_t(x_{t+1}).
$$

Let $x \in \mathcal{X}$ be arbitrary. Since $x_{T+1} \in \arg\min_{x \in \mathcal{X}} \sum_{t=0}^{T} g_t(x)$, we have

$$
\begin{aligned}
\sum_{t=0}^{T} g_t(x) &\geq \sum_{t=0}^{T} g_t(x_{T+1}) \\
&= \sum_{t=0}^{T-1} g_t(x_{T+1}) + g_T(x_{T+1}) \\
&\geq \sum_{t=0}^{T-1} g_t(x_{t+1}) + g_T(x_{T+1}) \qquad \text{by inductive hypothesis} \\
&= \sum_{t=0}^{T} g_t(x_{t+1}).
\end{aligned}
$$

This completes the proof. ∎

**Lemma B.2.** *For all $x \in \mathcal{X}$, FTRL (Eq. (4)) satisfies*

$$
\sum_{t=1}^{T} g_t(x_t) - \sum_{t=1}^{T} g_t(x) \leq \frac{D}{\eta} + \sum_{t=1}^{T} g_t(x_t) - g_t(x_{t+1}).
$$

*Proof.* Note that

$$
\sum_{t=1}^{T} g_t(x_t) - \sum_{t=1}^{T} g_t(x) \leq \sum_{t=1}^{T} g_t(x_t) - \sum_{t=1}^{T} g_t(x) + g_0(x) - g_0(x_1)
$$

because $x_1 \in \arg\min_{x \in \mathcal{X}} g_0(x)$. The proof of this lemma now follows by using the above inequality, Lemma B.1, the definition $g_0 = \frac{R}{\eta}$, and the definition of $D$. ∎

**Theorem B.1.** *FTRL (Eq. (4)) satisfies*

$$
\|x_{t+1} - x_t\|_{\mathcal{X}} \leq \eta \frac{L_g}{\alpha} \quad \text{and} \quad R_T^{static} \leq \frac{D}{\eta} + \eta \frac{T L_g^2}{\alpha}.
$$

*Choosing $\eta = \sqrt{\frac{\alpha D}{T L_g^2}}$ yields*

$$
R_T^{static} \leq O\left( \sqrt{\frac{D}{\alpha} T L_g^2} \right).
$$

*Proof.* Let $x^* \in \arg\min_{x \in \mathcal{X}} \sum_{t=1}^{T} g_t(x)$. Using Lemma B.2 we have

$$
\sum_{t=1}^{T} g_t(x_t) - \sum_{t=1}^{T} g_t(x^*) \leq \frac{D}{\eta} + \sum_{t=1}^{T} g_t(x_t) - g_t(x_{t+1}). \tag{5}
$$

We can bound the summands in the sum above as follows. Define $G_t(x) = \sum_{s=0}^{t-1} g_s(x)$. Then, $x_t \in \arg\min_{x \in \mathcal{X}} G_t(x)$. and $x_{t+1} \in \arg\min_{x \in \mathcal{X}} G_{t+1}(x)$. Since $\{g_s\}_{s=1}^{T}$ are convex, $R$ is $\alpha$-strongly-convex, and $g_0 = \frac{R}{\eta}$, we have that $G_t$ is $\frac{\alpha}{\eta}$-strongly-convex. So,

$$
G_t(x_{t+1}) \geq G_t(x_t) + \frac{\alpha}{2\eta} \|x_{t+1} - x_t\|_{\mathcal{X}}^2,
$$

$$
G_{t+1}(x_t) \geq G_{t+1}(x_{t+1}) + \frac{\alpha}{2\eta} \|x_{t+1} - x_t\|_{\mathcal{X}}^2.
$$

Adding the above two inequalities yields

$$
g_t(x_t) - g_t(x_{t+1}) \geq \frac{\alpha}{\eta} \|x_{t+1} - x_t\|_{\mathcal{X}}^2. \tag{6}
$$

Since $g_t$ are convex and $L_g$-Lipschitz continuous, we also have

$$g_t(x_t) - g_t(x_{t+1}) \leq L_g \|x_{t+1} - x_t\|_{\mathcal{X}}. \tag{7}$$

Combining Eqs. (6) and (7) we have

$$\|x_{t+1} - x_t\|_{\mathcal{X}} \leq \eta \frac{L_g}{\alpha}.$$

This proves the first part of the theorem. Now, using this in Eq. (7) we have

$$g_t(x_t) - g_t(x_{t+1}) \leq \eta \frac{L_g^2}{\alpha}. \tag{8}$$

Finally, substituting this in Eq. (5) proves the second part of the theorem. ∎

## C  Regret Analysis: Upper Bounds

First we prove a lemma that bounds the difference in the value of $f_t$ evaluated at the actual history $h_t$ and an idealized history that would have been obtained by playing $x_t$ in all prior rounds.

**Lemma C.1.** *Consider an online convex optimization with unbounded memory problem specified by* $(\mathcal{X}, \mathcal{H}, A, B)$. *If the decisions* $(x_t)$ *are generated by Algorithm 1, then*

$$\left| f_t(h_t) - \tilde{f}_t(x_t) \right| \leq \eta \frac{L\tilde{L}H_1}{\alpha}$$

*for all rounds* $t$. *When* $(\mathcal{X}, \mathcal{H}, A, B)$ *follows linear sequence dynamics with the* $\xi$-*weighted* $p$-*norm for* $p \geq 1$, *then*

$$\left| f_t(h_t) - \tilde{f}_t(x_t) \right| \leq \eta \frac{L\tilde{L}H_p}{\alpha}$$

*for all rounds* $t$.

*Proof.* We have

$$\left| f_t(h_t) - \tilde{f}_t(x_t) \right| = \left| f_t(h_t) - f_t\left(\sum_{k=0}^{t-1} A^k B x_t\right) \right| \qquad \text{by Definition 2.1}$$

$$\leq L \left\| h_t - \sum_{k=0}^{t-1} A^k B x_t \right\| \qquad \text{by Assumption A4}$$

$$= L \left\| \sum_{k=0}^{t-1} A^k B x_{t-k} - \sum_{k=0}^{t-1} A^k B x_t \right\| \qquad \text{by definition of } h_t$$

$$= L \underbrace{\left\| \sum_{k=0}^{t-1} A^k B (x_{t-k} - x_t) \right\|}_{(a)}. \tag{9}$$

First consider the general case where $(\mathcal{X}, \mathcal{H}, A, B)$ does not necessarily follow linear sequence dynamics. We can bound the term (a) as

$$\left\| \sum_{k=0}^{t-1} A^k B (x_{t-k} - x_t) \right\| \leq \sum_{k=0}^{t-1} \left\| A^k B \right\| \|x_t - x_{t-k}\|$$

$$\leq \sum_{k=0}^{t-1} \left\| A^k B \right\| k\eta \frac{\tilde{L}}{\alpha} \qquad \text{by Theorem B.1}$$

$$\leq \sum_{k=0}^{t-1} \left\| A^k \right\| k\eta \frac{\tilde{L}}{\alpha} \qquad \text{by Assumption A2}$$

$$\leq \eta \frac{\tilde{L}}{\alpha} H_1.$$

Plugging this into Eq. (9) completes the proof for the general case. Now consider the case when $(\mathcal{X}, \mathcal{H}, A, B)$ follows linear sequence dynamcis with the $\xi$-weighted $p$-norm. We can bound the term (a) as

$$\left\| \sum_{k=0}^{t-1} A^k B(x_{t-k} - x_t) \right\| = \| (0, A_0(x_t - x_{t-1}), A_1 A_0(x_t - x_{t-2}), \dots) \| \quad \text{by Definition 2.3}$$

$$= \left( \sum_{k=0}^{t-1} \xi_k^p \| A_{k-1} \cdots A_0 (x_t - x_{t-k}) \|^p \right)^{\frac{1}{p}} \quad \text{by Definition 2.3}$$

$$\leq \left( \sum_{k=0}^{t-1} \xi_k^p \| A_{k-1} \cdots A_0 \|^p \| x_t - x_{t-k} \|^p \right)^{\frac{1}{p}}$$

$$\leq \left( \sum_{k=0}^{t-1} \| A^k \|^p \| x_t - x_{t-k} \|^p \right)^{\frac{1}{p}} \quad \text{by Lemma A.1}$$

$$\leq \eta \frac{\tilde{L}}{\alpha} \left( \sum_{k=0}^{t-1} \| A^k \|^p k^p \right)^{\frac{1}{p}} \quad \text{by Theorem B.1}$$

$$\leq \eta \frac{\tilde{L}}{\alpha} H_p.$$

Plugging this into Eq. (9) completes the proof. ∎

Now we restate and prove Theorem 3.1

**Theorem 3.1.** *Consider an online convex optimization with unbounded memory problem specified by* $(\mathcal{X}, \mathcal{H}, A, B)$. *Let the regularizer* $R : \mathcal{X} \to \mathbb{R}$ *be* $\alpha$-*strongly-convex and satisfy* $|R(x) - R(\tilde{x})| \leq D$ *for all* $x, \tilde{x} \in \mathcal{X}$. *Algorithm 1 with step-size* $\eta$ *satisfies* $R_T(\text{FTRL}) \leq \frac{D}{\eta} + \eta \frac{T\tilde{L}^2}{\alpha} + \eta \frac{TL\tilde{L}H_1}{\alpha}$. *If* $\eta = \sqrt{\frac{\alpha D}{T\tilde{L}(LH_1 + \tilde{L})}}$, *then*

$$R_T(\text{FTRL}) \leq O\left( \sqrt{\frac{D}{\alpha} TL\tilde{L}H_1} \right).$$

*When* $(\mathcal{X}, \mathcal{H}, A, B)$ *follows linear sequence dynamics with the* $\xi$-*weighted* $p$-*norm, then all of the above hold with* $H_p$ *instead of* $H_1$.

*Proof.* First consider the general case where $(\mathcal{X}, \mathcal{H}, A, B)$ does not necessarily follow linear sequence dynamics. Let $x^* \in \arg\min_{x \in \mathcal{X}} \sum_{t=1}^{T} \tilde{f}_t(x)$. Note that we can write the regret as

$$R_T(\text{FTRL}) = \sum_{t=1}^{T} f_t(h_t) - \min_{x \in \mathcal{X}} \sum_{t=1}^{T} \tilde{f}_t(x)$$

$$= \underbrace{\sum_{t=1}^{T} f_t(h_t) - \tilde{f}_t(x_t)}_{(a)} + \underbrace{\sum_{t=1}^{T} \tilde{f}_t(x_t) - \tilde{f}_t(x^*)}_{(b)}.$$

We can bound term (a) using Lemma C.1 and term (b) using Theorem B.1. Therefore, we have

$$R_T(\text{FTRL}) = \underbrace{\sum_{t=1}^{T} f_t(h_t) - \tilde{f}_t(x_t)}_{(a)} + \underbrace{\sum_{t=1}^{T} \tilde{f}_t(x_t) - \tilde{f}_t(x^*)}_{(b)}$$

$$\leq \eta \frac{TL\tilde{L}H_1}{\alpha} + \frac{D}{\eta} + \eta \frac{T\tilde{L}^2}{\alpha}.$$

Choosing $\eta = \sqrt{\frac{\alpha D}{T\tilde{L}(LH_1+\tilde{L})}}$ yields

$$R_T(\texttt{FTRL}) \leq O\left(\sqrt{\frac{D}{\alpha}TL\tilde{L}H_1}\right),$$

where we used the definition of $p$-effective memory capacity (Definition 2.4) and the bound on $\tilde{L}$ (Theorem 2.1) to simplify the above expression. This completes the proof for the general case. The proof for when $(\mathcal{X}, \mathcal{H}, A, B)$ follows linear sequence dynamcis with the $\xi$-weighted $p$-norm is the same as above, except we bound the term (a) above using Lemma C.1 for linear sequence dynamics. ∎

Now we restate and prove Theorem 3.3.

**Theorem 3.3.** *Consider an online convex optimization with finite memory problem with constant memory length $m$ specified by $(\mathcal{X}, \mathcal{H} = \mathcal{X}^m, A_{finite,m}, B_{finite,m})$. Let the regularizer $R : \mathcal{X} \to \mathbb{R}$ be $\alpha$-strongly-convex and satisfy $|R(x) - R(\tilde{x})| \leq D$ for all $x, \tilde{x} \in \mathcal{X}$. Algorithm 1 with step-size $\eta = \sqrt{\frac{\alpha D}{T\tilde{L}(Lm^{\frac{3}{2}}+\tilde{L})}}$ satisfies*

$$R_T(\texttt{FTRL}) \leq O\left(\sqrt{\frac{D}{\alpha}TL\tilde{L}m^{\frac{3}{2}}}\right) \leq O\left(m\sqrt{\frac{D}{\alpha}TL^2}\right).$$

The OCO with finite memory problem, as defined in the literature, follows linear sequence dynamics with the 2-norm. In this subsection we consider a more general version of the OCO with finite memory problem that follows linear sequence dynamics with the $p$-norm. We provide an upper bound on the policy regret for this more general formulation and the proof of Theorem 3.3 follows as a special case when $p = 2$.

**Theorem C.1.** *Consider an online convex optimization with finite memory problem with constant memory length $m$, $(\mathcal{X}, \mathcal{H} = \mathcal{X}^m, A_{finite,m}, B_{finite,m})$. Assume that the problem follows linear sequence dynamics with the p-norm for $p \geq 1$. Let the regularizer $R : \mathcal{X} \to \mathbb{R}$ be $\alpha$-strongly-convex and satisfy $|R(x) - R(\tilde{x})| \leq D$ for all $x, \tilde{x} \in \mathcal{X}$. Algorithm 1 with step-size $\eta$ satisfies*

$$R_T(\texttt{FTRL}) \leq O\left(\sqrt{\frac{D}{\alpha}TL\tilde{L}m^{\frac{p+1}{p}}}\right) \leq O\left(\sqrt{\frac{D}{\alpha}TL^2m^{\frac{p+2}{p}}}\right).$$

*Proof.* Using Theorem 3.1 it suffices to bound $\tilde{L}$ and $H_p$ for this problem. Note that $\|A_{finite}^k\| = 1$ if $k \leq m$ and 0 otherwise. Using this we have

$$H_p = \left(\sum_{k=0}^{\infty}\left(k\|A_{finite}^k\|\right)^p\right)^{\frac{1}{p}} = \left(\sum_{k=0}^{m}k^p\right)^{\frac{1}{p}} \leq O\left(m^{\frac{p+1}{p}}\right).$$

This proves the first inequality in the statement of the theorem. The second inequality follows from the above and Theorem 2.1, which states that

$$\tilde{L} \leq L\left(\sum_{k=0}^{\infty}\|A_{finite}^k\|^p\right)^{\frac{1}{p}} = Lm^{\frac{1}{p}}. \qquad \blacksquare$$

Finally, we provide an upper bound on the policy regret for the OCO with $\rho$-discounted infinite memory problem. For simplicity, we consider the case when the problem follows linear sequence dynamics with the 2-norm instead of a general $p$-norm.

**Theorem C.2.** *Consider an online convex optimization with $\rho$-discounted infinite memory problem $(\mathcal{X}, \mathcal{H}, A_{infinite,\rho}, B_{infinite})$. Suppose that the problem follows linear sequence dynamics with the 2-norm. Let the regularizer $R : \mathcal{X} \to \mathbb{R}$ be $\alpha$-strongly-convex and satisfy $|R(x) - R(\tilde{x})| \leq D$ for all $x, \tilde{x} \in \mathcal{X}$. Algorithm 1 with step-size $\eta$ satisfies*

$$R_T(\texttt{FTRL}) \leq O\left(\sqrt{\frac{D}{\alpha}TL\tilde{L}(1-\rho^2)^{-\frac{3}{2}}}\right) \leq O\left(\sqrt{\frac{D}{\alpha}TL^2(1-\rho^2)^{-2}}\right) \leq O\left(\sqrt{\frac{D}{\alpha}TL^2(1-\rho)^{-2}}\right).$$

*Proof.* Using Theorem 3.1, it suffices to bound $\tilde{L}$ and $H_p$ for this problem. Recall that $\|A_{\text{infinite},\rho}^k\| = \rho^k$. Using this we have

$$H_2 = \left(\sum_{k=0}^{\infty} \left(k\|A_{\text{finite}}^k\|\right)^2\right)^{\frac{1}{2}} = \left(\sum_{k=0}^{\infty} \left(k\rho^k\right)^2\right)^{\frac{1}{2}} \leq (1-\rho^2)^{-\frac{3}{2}}.$$

This proves the first inequality in the statement of the theorem. The second inequality follows from the above and Theorem 2.1, which states that

$$\tilde{L} \leq L\left(\sum_{k=0}^{\infty} \|A_{\text{infinite},\rho}^k\|^2\right)^{\frac{1}{2}} = L(1-\rho^2)^{-\frac{1}{2}}.$$

The last inequality follows because $1 - \rho^2 = (1+\rho)(1-\rho)$, which implies that $1 - \rho \leq 1 - \rho^2 \leq 2(1-\rho)$ because $\rho \in (0,1)$. ∎

## C.1 Existing Regret Bound for OCO with Finite Memory

In this subsection we provide a detailed comparison of our upper bound on the policy regret for OCO with finite memory with that of Anava et al. [2015]. The material in this subsection comes from Appendix A.2 of their arXiv version or Appendix C.2 of their conference version.

The existing upper bound on regret is

$$O\left(\sqrt{DT\lambda m^{\frac{3}{2}}}\right),$$

where $D = \max_{x,\tilde{x} \in \mathcal{X}} |R(x) - R(\tilde{x})|$. Although the parameter $\lambda$ is defined in terms of dual norms of the gradient of $\tilde{f}_t$, it is essentially the Lipschitz-continuity constant for $\tilde{f}_t$: for all $x, \tilde{x} \in \mathcal{X}$,

$$\left|\tilde{f}_t(x) - \tilde{f}_t(\tilde{x})\right| \leq \sqrt{\lambda\alpha}\|x - \tilde{x}\|,$$

where $\alpha$ is the strong-convexity parameter of the regularizer $R$ (or $\sigma$ in the notation of Anava et al. [2015]). Therefore, the existing regret bound can be rewritten as

$$O\left(\tilde{L}\sqrt{\frac{D}{\alpha}Tm^{\frac{3}{2}}}\right).$$

Our upper bound on the policy regret for OCO with finite memory Theorem 3.3 is

$$O\left(\sqrt{\frac{D}{\alpha}L\tilde{L}Tm^{\frac{3}{2}}}\right).$$

Since $\tilde{L} \leq \sqrt{m}L$ by Theorem 2.1, this leads to an improvement by a factor of $m^{\frac{1}{4}}$.

# D   Regret Analysis: Lower Bounds

We first restate Theorems 3.2 and 3.4.

**Theorem 3.2.** *There exists an instance of the online convex optimization with unbounded memory problem, $(\mathcal{X}, \mathcal{H}, A, B)$, that follows linear sequence dynamics with the $\xi$-weighted $p$-norm and there exist $L$-Lipschitz continuous loss functions $\{f_t : \mathcal{H} \to \mathbb{R}\}_{t=1}^T$ such that the regret of any algorithm $\mathcal{A}$ satisfies*

$$R_T(\mathcal{A}) \geq \Omega\left(\sqrt{TL\tilde{L}H_p}\right).$$

**Theorem 3.4.** *There exists an instance of the online convex optimization with finite memory problem with constant memory length $m$, $(\mathcal{X}, \mathcal{H} = \mathcal{X}^m, A_{\text{finite},m}, B_{\text{finite},m})$, and there exist $L$-Lipschitz continuous loss functions $\{f_t : \mathcal{H} \to \mathbb{R}\}_{t=1}^T$ such that the regret of any algorithm $\mathcal{A}$ satisfies*

$$R_T(\mathcal{A}) \geq \Omega\left(m\sqrt{TL^2}\right).$$

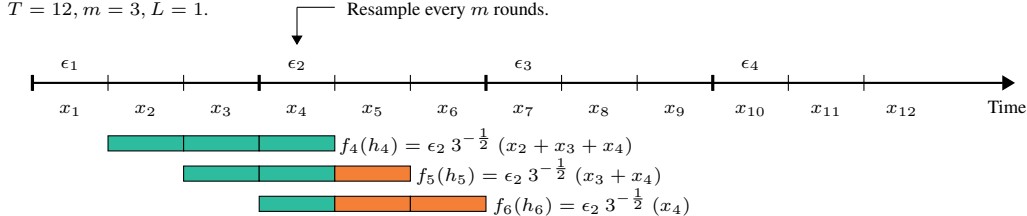

Figure 2: An illustration of the loss functions $f_t$ for the OCO with finite memory lower bound. Suppose $T = 12, m = 3, L = 1$, and $p = 2$. Time is divided into blocks of size $m = 3$. Consider round $t = 5$. The history is $h_5 = (x_3, x_4, x_5)$. The loss function $f_5(h_5)$ is a product of three terms: a random sign $\epsilon_2$ sampled for the block that round 5 belongs to, namely, block 2; a scaling factor of $m^{-\frac{1}{2}}$; a sum over the decisions in the history excluding those that were chosen after observing $\epsilon_2$, i.e., a sum over $x_3$ and $x_4$, excluding $x_5$.

Theorem 3.2 follows from Theorem 3.4. However, the lower bound is true for a much broader class of problems as we show in this section. We first provide a lower bound for a more general formulation of the OCO with finite memory problem (Theorem D.1). The proof of Theorem 3.4 follows as a special case when $p = 2$. Then, we provide a lower bound for the OCO with $\rho$-discounted infinite memory problem (Theorem D.2).

The OCO with finite memory problem, as defined in the literature, follows linear sequence dynamics with the 2-norm. In this section we consider a more general version of the OCO with finite memory problem that follows linear sequence dynamics with the $p$-norm. We provide a lower bound on the policy regret for this more general formulation and the proof of Theorem 3.4 follows as a special case when $p = 2$.

**Theorem D.1.** *For all $p \geq 1$, there exists an instance of the online convex optimization with finite memory problem with constant memory length $m$, $(\mathcal{X}, \mathcal{H} = \mathcal{X}^m, A_{\text{finite},m}, B_{\text{finite},m})$, that follows linear sequence dynamics with the $p$-norm, and there exist $L$-Lipschitz continuous loss functions $\{f_t : \mathcal{H} \to \mathbb{R}\}_{t=1}^{T}$ such that the regret of any algorithm $\mathcal{A}$ satisfies*

$$R_T(\mathcal{A}) \geq \Omega\left(\sqrt{TL^2 m^{\frac{p+2}{p}}}\right).$$

*Proof.* Let $\mathcal{X} = [-1, 1]$ and consider an OCO with finite memory problem with constant memory length $m$, $(\mathcal{X}, \mathcal{H} = \mathcal{X}^m, A_{\text{finite},m}, B_{\text{finite},m})$, that follows linear sequence dynamics with the $p$-norm. For simplicity, assume that $T$ is a multiple of $m$ (otherwise, the same proof works but with slightly more tedious bookkeeping) and that $L = 1$ (otherwise, multiply the functions $f_t$ defined below by $L$).

Divide the $T$ rounds into $N = \frac{T}{m}$ blocks of $m$ rounds each. Sample $N$ independent Rademacher random variables $\{\epsilon_1, \ldots, \epsilon_N\}$, where each $\epsilon_i$ is equal to $\pm 1$ with probability $\frac{1}{2}$. Recall that $h_t = (x_t, \ldots, x_{t-m+1})$. Define the loss functions $\{f_t\}_{t=1}^{T}$ as follows. (See Fig. 2 for an illustration.) If $t \leq m$, let $f_t = 0$. Otherwise, let

$$f_t(h_t) = \epsilon_{\lceil \frac{t}{m} \rceil} m^{\frac{1-p}{p}} \sum_{k=0}^{m-1-(t-m\lfloor \frac{t}{m} \rfloor - 1)} x_{m\lfloor \frac{t}{m} \rfloor + 1 - k}$$

$$= \epsilon_{\lceil \frac{t}{m} \rceil} m^{\frac{1-p}{p}} \left( x_{t-m+1} + \cdots + x_{m\lfloor \frac{t}{m} \rfloor + 1} \right).$$

In words, the loss in the first $m$ rounds is equal to 0. Thereafter, in round $t$ the loss is equal to a random sign $\epsilon_{\lceil \frac{t}{m} \rceil}$, which is *fixed for that block*, times a scaling factor, which is chosen according to the $p$-norm to ensure that the Lipschitz constant $L$ is at most 1, times a sum of a *subset* of past decisions in the history $h_t = (x_t, \ldots, x_{t-m+1})$. This subset consists of all past decisions until and including the first decision of the current block, which is the decision in round $m\lfloor \frac{t}{m} \rfloor + 1$.

The functions $f_t$ are linear, so they are convex. In order to show that they satisfy Assumptions **A3** and **A4**, it remains to show that they are 1-Lipschitz continuous. Let $h = (x^{(1)}, \ldots, x^{(m)})$ and

$\tilde{h} = (\tilde{x}^{(1)}, \ldots, \tilde{x}^{(m)})$ be arbitrary elements of $\mathcal{H} = \mathcal{X}^m$. We have

$$\left| f_t(h) - f_t(\tilde{h}) \right|$$

$$\leq \left| \epsilon_{\lceil \frac{t}{m} \rceil} m^{\frac{1-p}{p}} \left( (x^{(1)} - \tilde{x}^{(1)}) + \cdots + (x^{(m)} - \tilde{x}^{(m)}) \right) \right|$$

$$\leq m^{\frac{1-p}{p}} \left| (x^{(1)} - \tilde{x}^{(1)}) + \cdots + (x^{(m)} - \tilde{x}^{(m)}) \right| \qquad \text{because } \epsilon_{\lceil \frac{t}{m} \rceil} \in \{-1, +1\}$$

$$\leq m^{\frac{1-p}{p}} m^{1 - \frac{1}{p}} \left( \sum_{k=1}^{m} \left| x^{(k)} - \tilde{x}^{(k)} \right|^p \right)^{\frac{1}{p}} \qquad \text{by Hölder's inequality}$$

$$= \|h - \tilde{h}\|_{\mathcal{H}},$$

where the last equality follows because of our assumption that the problem that follows linear sequence dynamics with the $p$-norm.

First we will show that the total expected loss of any algorithm is 0, where the expectation is with respect to the randomness in the choice of $\{\epsilon_1, \ldots, \epsilon_N\}$. The total loss in the first block is 0 because $f_t = 0$ for $t \in [m]$. For each subsequent block $n \in \{2, \ldots, N\}$, the total loss in block $n$ depends on the algorithm's choices made *before* observing $\epsilon_n$, namely, $\{x_{(n-2)m+2}, \ldots, x_{(n-1)m+1}\}$. Since $\epsilon_n$ is equal to $\pm 1$ with probability $\frac{1}{2}$, the expected loss of any algorithm in a block is equal to 0 and the total expected loss is also equal to 0.

Now we will show that the expected loss of the benchmark is at most

$$-O\left( \sqrt{Tm^{\frac{p+2}{p}}} \right),$$

where the expectation is with respect to the randomness in the choice of $\{\epsilon_1, \ldots, \epsilon_N\}$. We have

$$\mathbb{E}\left[ \min_{x \in \mathcal{X}} \sum_{t=1}^{T} \tilde{f}_t(x) \right] = \mathbb{E}\left[ \min_{x \in \mathcal{X}} \sum_{n=2}^{N} \sum_{t=(n-1)m+1}^{nm} \tilde{f}_t(x) \right]$$

$$= \mathbb{E}\left[ \min_{x \in \mathcal{X}} \sum_{n=2}^{N} \sum_{t=(n-1)m+1}^{nm} \epsilon_n m^{\frac{1-p}{p}} \times x \times (m - (t - (n-1)m - 1)) \right].$$

The first equality follows from first summing over blocks and then summing over the rounds in that block. The second equality follows from the definitions of $f_t$ above and of $\tilde{f}_t$ (Definition 2.1). By the defintion of $\tilde{f}_t$, the history $h_t$ consists of $m$ copies of $x$ for $t \geq m$.. By the definition of $f_t$, which sums over all past decisions until the first round of the current block, we have that within a block the sum first extends over $m$ copies of $x$ (in the first round of the block), then $m - 1$ copies of $x$ (in the second round of the block), and so on until the last round of the block. So, we have

$$\mathbb{E}\left[ \min_{x \in \mathcal{X}} \sum_{t=1}^{T} \tilde{f}_t(x) \right] = \mathbb{E}\left[ \min_{x \in \mathcal{X}} \sum_{n=2}^{N} \sum_{t=(n-1)m+1}^{nm} \epsilon_n m^{\frac{1-p}{p}} \times x \times (m - (t - (n-1)m - 1)) \right]$$

$$= \mathbb{E}\left[ \min_{x \in \mathcal{X}} \sum_{n=2}^{N} \sum_{k=0}^{m-1} \epsilon_n m^{\frac{1-p}{p}} \times x \times (m - k) \right]$$

$$= m^{\frac{1-p}{p}} \frac{m^2 + m}{2} \mathbb{E}\left[ \min_{x \in \mathcal{X}} \sum_{n=2}^{N} \epsilon_n x \right]$$

$$= m^{\frac{1-p}{p}} \frac{m^2 + m}{2} \mathbb{E}\left[ \min_{x \in \{-1,1\}} \sum_{n=2}^{N} \epsilon_n x \right]$$

$$= m^{\frac{1-p}{p}} \frac{m^2 + m}{2} \mathbb{E}\left[ \frac{1}{2} \sum_{n=2}^{N} \epsilon_n(-1 + 1) - \frac{1}{2} \left| \sum_{n=2}^{N} \epsilon_n(-1 - 1) \right| \right],$$

where the second-last equality follows because the minima of a linear function over an interval is at one of the endpoints and the last equality follows because $\min\{x, y\} = \frac{1}{2}(x+y) - \frac{1}{2}|x-y|$. Since $\epsilon_n$ are Rademacher random variables equal to $\pm 1$ with probability $\frac{1}{2}$, we can simplify the above as

$$\mathbb{E}\left[\min_{x \in \mathcal{X}} \sum_{t=1}^{T} \tilde{f}_t(x)\right] = m^{\frac{1-p}{p}} \frac{m^2 + m}{2} \mathbb{E}\left[-\frac{1}{2}\left|\sum_{n=2}^{N} -2\epsilon_n\right|\right]$$

$$= m^{\frac{1-p}{p}} \frac{m^2 + m}{2} \mathbb{E}\left[-\left|\sum_{n=2}^{N} \epsilon_n\right|\right]$$

$$= -m^{\frac{1-p}{p}} \frac{m^2 + m}{2} \mathbb{E}\left[\left|\sum_{n=2}^{N} \epsilon_n\right|\right]$$

$$\leq -m^{\frac{1-p}{p}} \frac{m^2 + m}{2} \sqrt{N},$$

where the last inequality follows from Khintchine's inequality. Using the definition $N = \frac{T}{m}$, we have

$$\mathbb{E}\left[\min_{x \in \mathcal{X}} \sum_{t=1}^{T} \tilde{f}_t(x)\right] \leq -m^{\frac{1-p}{p}} \frac{m^2 + m}{2} \sqrt{\frac{T}{m}}$$

$$= -\frac{1}{2}\sqrt{T}\left(m^{\frac{3}{2} + \frac{1-p}{p}} + m^{\frac{1}{2} + \frac{1-p}{p}}\right)$$

$$\leq -O\left(\sqrt{T} m^{\frac{3}{2} + \frac{1-p}{p}}\right)$$

$$= -O\left(\sqrt{T} m^{\frac{p+2}{2p}}\right)$$

$$= -O\left(\sqrt{T m^{\frac{p+2}{p}}}\right).$$

Therefore, we have

$$\mathbb{E}_{\epsilon_1, \ldots, \epsilon_N}\left[R_T(\texttt{FTRL})\right] = \mathbb{E}\left[\sum_{t=1}^{T} f_t(h_t)\right] - \mathbb{E}\left[\min_{x \in \mathcal{X}} \sum_{t=1}^{T} \tilde{f}_t(x)\right] \geq \Omega\left(\sqrt{T m^{\frac{p+2}{p}}}\right).$$

This completes the proof. ∎

Now we provide a lower bound for the OCO with $\rho$-discounted infinite memory problem. For simplicity, we consider the case when the problem follows linear sequence dynamics with the 2-norm instead of a general $p$-norm.

**Theorem D.2.** *Let $\rho \in [\frac{1}{2}, 1)$. There exists an instance of the online convex optimization with $\rho$-discounted infinite memory problem, $(\mathcal{X}, \mathcal{H}, A_{\text{infinite},\rho}, B_{\text{infinite}})$, that follows linear sequence dynamics with the 2-norm and there exist L-Lipschitz continuous loss functions $\{f_t : \mathcal{H} \to \mathbb{R}\}_{t=1}^{T}$ such that the regret of any algorithm $\mathcal{A}$ satisfies*

$$R_T(\mathcal{A}) \geq \Omega\left(\sqrt{T L^2 (1-\rho)^{-2}}\right).$$

The proof is very similar to that of Theorem D.1 with slight adjustments to account for a $\rho$-discounted infinite memory instead of a finite memory of constant size $m$.

*Proof.* Let $\mathcal{X} = [-1, 1]$ and consider an OCO with infinite memory problem with discount factor $\rho$, $(\mathcal{X}, \mathcal{H}, A_{\text{infinite},\rho}, B_{\text{infinite}})$, that follows linear sequence dynamics with the 2-norm. For simplicity, assume that $T$ is a multiple of $(1-\rho)^{-1}$ (otherwise, the same proof works but with slightly more tedious bookkeeping) and that $L = 1$ (otherwise, multiply the functions $f_t$ defined below by $L$).

Define $m = (1-\rho)^{-1}$. Divide the $T$ rounds into $N = \frac{T}{m}$ blocks of $m$ rounds each. Sample $N$ independent Rademacher random variables $\{\epsilon_1, \ldots, \epsilon_N\}$, where each $\epsilon_i$ is equal to $\pm 1$ with

probability $\frac{1}{2}$. Recall that $h_t = (x_t, \rho x_{t-1}, \ldots, \rho^{t-1} x_1, 0, \ldots)$. Define the loss functions $\{f_t\}_{t=1}^T$ as follows. If $t \leq m$, let $f_t = 0$. Otherwise, let

$$f_t(h_t) = \epsilon_{\lceil \frac{t}{m} \rceil} m^{-\frac{1}{2}} \sum_{k=0}^{m-1} \rho^{k+t-m\lfloor \frac{t}{m} \rfloor - 1} x_{m\lfloor \frac{t}{m} \rfloor + 1 - k}.$$

The functions $f_t$ are linear, so they are convex. In order to show that they satisfy Assumptions **A3** and **A4**, it remains to show that they are 1-Lipschitz continuous. Let $h = (x^{(1)}, \rho x^{(2)}, \ldots)$ and $\tilde{h} = (\tilde{x}^{(1)}, \rho \tilde{x}^{(2)}, \ldots)$ be arbitrary elements of $\mathcal{H}$. We have

$$\left| f_t(h) - f_t(\tilde{h}) \right|$$

$$\leq \left| \epsilon_{\lceil \frac{t}{m} \rceil} m^{-\frac{1}{2}} \sum_{k=1}^{m} \rho^{k-1} \left( x^{(k)} - \tilde{x}^{(k)} \right) \right|$$

$$\leq m^{-\frac{1}{2}} \left| \sum_{k=1}^{m} \rho^{k-1} \left( x^{(k)} - \tilde{x}^{(k)} \right) \right| \qquad \text{because } \epsilon_{\lceil \frac{t}{m} \rceil} \in \{-1, +1\}$$

$$\leq m^{-\frac{1}{2}} m^{\frac{1}{2}} \left( \sum_{k=1}^{m} \rho^{2(k-1)} \left| x^{(k)} - \tilde{x}^{(k)} \right|^2 \right)^{\frac{1}{2}} \qquad \text{by Hölder's inequality}$$

$$\leq \|h - \tilde{h}\|_{\mathcal{H}},$$

where the last equality follows because the follows linear sequence dynamics with the 2-norm.

First we will show that the total expected loss of any algorithm is 0, where the expectation is with respect to the randomness in the choice of $\{\epsilon_1, \ldots, \epsilon_N\}$. The total loss in the first block is 0 because $f_t = 0$ for $t \in [m]$. For each subsequent block $n \in \{2, \ldots, N\}$, the total loss in block $n$ depends on the algorithm's choices made *before* observing $\epsilon_n$, namely, $\{x_{(n-2)m+2}, \ldots, x_{(n-1)m+1}\}$. Since $\epsilon_n$ is equal to $\pm 1$ with probability $\frac{1}{2}$, the expected loss of any algorithm in a block is equal to 0 and the total expected loss is also equal to 0.

Now we will show that the expected loss of the benchmark is at most

$$-O\left( \sqrt{T(1-\rho)^{-2}} \right),$$

where the expectation is with respect to the randomness in the choice of $\{\epsilon_1, \ldots, \epsilon_N\}$. We have

$$\mathbb{E}\left[ \min_{x \in \mathcal{X}} \sum_{t=1}^{T} \tilde{f}_t(x) \right] = \mathbb{E}\left[ \min_{x \in \mathcal{X}} \sum_{n=2}^{N} \sum_{t=(n-1)m+1}^{nm} \tilde{f}_t(x) \right]$$

$$= \mathbb{E}\left[ \min_{x \in \mathcal{X}} \sum_{n=2}^{N} \sum_{t=(n-1)m+1}^{nm} \epsilon_n m^{-\frac{1}{2}} \sum_{k=0}^{m-1} \rho^{k+t-(n-1)m-1} x \right]$$

$$= m^{-\frac{1}{2}} \mathbb{E}\left[ \min_{x \in \mathcal{X}} \sum_{n=2}^{N} \epsilon_n x \sum_{t=(n-1)m+1}^{nm} \rho^{t-(n-1)m-1} \sum_{k=0}^{m-1} \rho^k \right]$$

$$= m^{-\frac{1}{2}} \frac{1-\rho^m}{1-\rho} \mathbb{E}\left[ \min_{x \in \mathcal{X}} \sum_{n=2}^{N} \epsilon_n x \sum_{t=(n-1)m+1}^{nm} \rho^{t-(n-1)m-1} \right]$$

$$= m^{-\frac{1}{2}} \left( \frac{1-\rho^m}{1-\rho} \right)^2 \underbrace{\mathbb{E}\left[ \min_{x \in \mathcal{X}} \sum_{n=2}^{N} \epsilon_n x \right]}_{(a)}.$$

The term (a) above can be bounded above by $-\sqrt{N}$ as in the proof of Theorem D.1 using Khintchine's inequality. Therefore, using that $N = \frac{T}{m}$ and $m = (1-\rho)^{-1}$ we have

$$\mathbb{E}\left[\min_{x\in\mathcal{X}}\sum_{t=1}^{T}\tilde{f}_t(x)\right] \le -m^{-\frac{1}{2}}\left(\frac{1-\rho^m}{1-\rho}\right)^2\sqrt{N}$$

$$\le -(1-\rho)^{\frac{1}{2}}\left(\frac{1-\rho^m}{1-\rho}\right)^2\sqrt{T(1-\rho)}$$

$$= -\sqrt{T}\frac{(1-\rho^m)^2}{1-\rho}$$

$$= -\sqrt{T(1-\rho)^{-2}}(1-\rho^m)^2$$

$$\le -O\left(\sqrt{T(1-\rho)^{-2}}\right),$$

where the last inequality follows from the assumption that $\rho \in [\frac{1}{2}, 1)$ and the following argument:

$$\rho^m = (1 - (1-\rho))^m = (1 - (1-\rho))^{\frac{1}{1-\rho}} \le \frac{1}{e}$$

$$\Rightarrow (1-\rho^m) \ge 1 - \frac{1}{e}$$

$$\Rightarrow (1-\rho^m)^2 \ge \left(1 - \frac{1}{e}\right)^2$$

$$\Rightarrow -(1-\rho^m)^2 \le -\left(1 - \frac{1}{e}\right)^2$$

This completes the proof. ∎

## E  Online Linear Control

### E.1  Formulation as OCO with Unbounded Memory

Now we formulate the online linear control problem in our framework by defining the decision space $\mathcal{X}$, the history space $\mathcal{H}$, and the linear operators $A : \mathcal{H} \to \mathcal{H}$ and $B : \mathcal{W} \to \mathcal{H}$. Then, we define the functions $f_t : \mathcal{H} \to \mathbb{R}$ in terms of $c_t$ and finally, prove an upper bound on the policy regret. For notational convenience, let $(M^{[s]})$ and $(Y_k)$ denote the sequences $(M^{[1]}, M^{[2]}, \dots)$ and $(Y_0, Y_1, \dots)$ respectively.

Recall that we fix $K \in \mathcal{K}$ to be an arbitrary $(\kappa, \rho)$-strongly stable linear controller and consider the disturbance-action controller policy class $\mathcal{M}_K$ (Definition 4.1). For the rest of this paper let $\widetilde{F} = F - GK$. The first step is a change of variables with respect to the control inputs from linear controllers to DACs and the second is a corresponding change of variables for the state. Define the decision space $\mathcal{X}$ as

$$\mathcal{X} = \{M = (M^{[s]}) : M^{[s]} \in \mathbb{R}^{d\times d}, \|M^{[s]}\|_2 \le \kappa^4\rho^s\} \tag{10}$$

with

$$\|M\|_{\mathcal{X}} = \sqrt{\sum_{s=1}^{\infty}\rho^{-s}\|M^{[s]}\|_F^2}. \tag{11}$$

Define the history space $\mathcal{H}$ to be the set consisting of sequences $h = (Y_k)$, where $Y_0 \in \mathcal{X}$ and $Y_k = \widetilde{F}^{k-1}GX_k$ for $X_k \in \mathcal{X}, k \ge 1$ with

$$\|h\|_{\mathcal{H}} = \sqrt{\sum_{k=0}^{\infty}\xi_k^2\|Y_k\|_{\mathcal{X}}^2}, \tag{12}$$

where the weights $(\xi_k)$ are nonnegative real numbers defined as

$$\xi = (1, 1, 1, \rho^{-\frac{1}{2}}, \rho^{-1}, \rho^{-\frac{3}{2}}, \dots). \tag{13}$$

Define the linear operators $A : \mathcal{H} \to \mathcal{H}$ and $B : \mathcal{W} \to \mathcal{H}$ as

$$A((Y_0, Y_1, \dots)) = (0, GY_0, \widetilde{F}Y_1, \widetilde{F}Y_2, \dots) \quad \text{and} \quad B(M) = (M, 0, 0, \dots).$$

Note that the problem follows linear sequence dynamics with the $\xi$-weighted 2-norm (Definition 2.3), where $\xi$ is defined above in Eq. (13). The weights in the weighted norms on $\mathcal{X}$ and $\mathcal{H}$ increase exponentially. However, the norms $\|M^{[s]}\|_F^2$ and $\|\widetilde{F}^{k-1}G\|_F^2$ decrease exponentially as well: by definition of $M^{[s]}$ in Eq. (11) and the assumption on $\widetilde{F} = F - GK$ for $K \in \mathcal{K}$. Leveraging this exponential decrease in $\|M^{[s]}\|_F^2$ and $\|\widetilde{F}^{k-1}G\|_F^2$ to define exponentially increasing weights turns out to be crucial for deriving our regret bounds that are stronger than existing results. Furthermore, the choice to have $\xi_p = 1$ for $p \in \{1, 2\}$ in addition to $p = 0$ (as required by Definition 2.3) might seem like a small detail, but this also turns out to be crucial for avoiding unnecessary factors of $\rho^{-1}$ in the regret bounds.

Recall that the loss functions in the online linear control problem are $c_t(s_t, u_t)$, where $s_t$ and $u_t$ are the state and control at round $t$. Now we will show how to construct the functions $f_t : \mathcal{H} \to \mathbb{R}$ that correspond to $c_t(s_t, u_t)$. By definition, given a sequence of decisions $(M_0, \dots, M_t)$, the history at the end of round $t$ is given by

$$h_t = (M_t, GM_{t-1}, \widetilde{F}GM_{t-2}, \dots, \widetilde{F}^{t-1}GM_0, 0, \dots).$$

A simple inductive argument shows that the state and control in round $t$ can be written as

$$s_t = \widetilde{F}^t s_0 + \sum_{k=0}^{t-1} \sum_{s=1}^{k+1} \widetilde{F}^{t-k-1} GM_k^{[s]} w_{k-s} + w_{t-1}, \tag{14}$$

$$u_t = -Ks_t + \sum_{s=1}^{t+1} M_t^{[s]} w_{t-s}. \tag{15}$$

Define the functions $f_t : \mathcal{H} \to \mathbb{R}$ by $f_t(h) = c_t(s, u)$, where $s$ and $u$ are the state and control determined by the history as above. Note that $f_t$ is parameterized by the past disturbances. Since the state and control are linear functions of the history and $c_t$ is convex, this implies that $f_t$ is convex.

With the above formulation and the fact that the class of disturbance-action controllers is a superset of the class of $(\kappa, \rho)$-strongly-stable linear controllers, we have that the policy regret for the online linear control problem is at most

$$\sum_{t=0}^{T-1} f_t(h_t) - \min_{M \in \mathcal{X}} \sum_{t=0}^{T-1} \tilde{f}_t(M).$$

This completes the specification of the online convex optimization with unbounded memory problem, $(\mathcal{X}, \mathcal{H}, A, B)$, corresponding to the online linear control problem. Using Algorithm 1 and Theorem 3.1 we can upper bound the above by

$$O\left( \sqrt{\frac{D}{\alpha} TL\tilde{L}H_2} \right),$$

where $L$ is the Lipschitz constant of $f_t$, $\tilde{L}$ is the Lipschitz constant of $\tilde{f}_t$, $H_2$ is the 2-effective memory capacity, and $D = \max_{x, \tilde{x} \in \mathcal{X}} |R(x) - R(\tilde{x})|$ for an $\alpha$-strongly-convex regularizer $R : \mathcal{X} \to \mathbb{R}$. In the next subsection we bound these quantities in terms of the problem parameters of the online linear control problem. We use $O(\cdot)$ to hide absolute constants.

## E.2 Regret Analysis

We use the following standard facts about matrix norms.

**Lemma E.1.** *Let* $M, N \in \mathbb{R}^{d \times d}$. *Then,*

1. $\|M\|_2 \leq \|M\|_F \leq \sqrt{d}\|M\|_2$.

2. $\|MN\|_F \leq \|M\|_2\|N\|_F$.

*Proof.* Part 1 can be found in, for example, Golub and Loan [1996, Section 2.3.2]. Letting $N_j$ denote the $j$-th column of $N$, part 2 follows from

$$\|MN\|_F^2 = \sum_{j=1}^d \|MN_j\|_2^2 \leq \|M\|_2^2 \sum_{j=1}^d \|N_j\|_2^2 = \|M\|_2^2\|N\|_F^2.$$

This completes the proof. ∎

**Lemma E.2.** *For $s \geq 2$, the operator norm $\|A^s\|$ is bounded above as*

$$\|A^s\| \leq O\left(\kappa^4 \rho^{\frac{s}{2}}\right).$$

*Proof.* Recall the definition of $\mathcal{H}$ and $\|\cdot\|_{\mathcal{H}}$ (Eq. (12)). Let

$$(Y_0, Y_1, \dots) = (Y_0, GX_1, \widetilde{F}GX_2, \widetilde{F}^2GX_3, \dots)$$

be an element of $\mathcal{H}$ with unit norm, i.e.,

$$\sqrt{\sum_{k=0}^\infty \xi_k^2\|Y_k\|_{\mathcal{X}}^2} = 1,$$

where the weights $(\xi_k)$ are defined in Eq. (13). Note that $\xi_p = 1$ for $p = 0, 1$ and $\xi_p^2 = \rho^{-p+2}$ for $p = 2, 3, \dots$. From the definition of the operator $A$ and for $s \geq 2$, we have

$$A^s((Y_0, Y_1, \dots)) = (0, \dots, 0, \widetilde{F}^{s-1}GY_0, \widetilde{F}^sGX_1, \widetilde{F}^{s+1}GX_2, \dots).$$

Now we bound $\|A^s\|$ as follows. By definition of $A^s$ and $\|\cdot\|_{\mathcal{H}}$ (Eq. (12)), and part 2 of Lemma E.1, we have

$$\|A^s((Y_0, Y_1, \dots))\| = \sqrt{\rho^{-s+2}\|\widetilde{F}^{s-1}GY_0\|_{\mathcal{X}}^2 + \sum_{k=1}^\infty \rho^{-s-k+2}\|\widetilde{F}^{s+k-1}GX_k\|_{\mathcal{X}}^2}$$

$$\leq \sqrt{\rho^{-s+2}\|\widetilde{F}^{s-1}G\|_2^2\|Y_0\|_{\mathcal{X}}^2 + \sum_{k=1}^\infty \rho^{-s-k+2}\|\widetilde{F}^{s-1}\|_2^2\|\widetilde{F}\|_2^2\|\widetilde{F}^{k-1}GX_k\|_{\mathcal{X}}^2}$$

$$\leq \rho^{-\frac{s}{2}}\|\widetilde{F}^{s-1}\|_2\sqrt{\rho^2\|G\|_2^2\|Y_0\|_{\mathcal{X}}^2 + \sum_{k=1}^\infty \rho^{-k+2}\|\widetilde{F}\|_2^2\|\widetilde{F}^{k-1}GX_k\|_{\mathcal{X}}^2}$$

$$= \rho^{-\frac{s}{2}}\|\widetilde{F}^{s-1}\|_2\sqrt{\rho^2\|G\|_2^2\|Y_0\|_{\mathcal{X}}^2 + \sum_{k=1}^\infty \rho^{-k+2}\|\widetilde{F}\|_2^2\|Y_k\|_{\mathcal{X}}^2}.$$

Using our assumptions that $\|G\|_2 \leq \kappa$ and $\|\widetilde{F}\|_2 \leq \kappa^2\rho$, we have

$$\|A^s((Y_0, Y_1, \dots))\| \leq \rho^{-\frac{s}{2}}\|\widetilde{F}^{s-1}\|_2\sqrt{\rho^2\kappa^2\|Y_0\|_{\mathcal{X}}^2 + \sum_{k=1}^\infty \rho^{-k+2}\kappa^4\rho^2\|Y_k\|_{\mathcal{X}}^2}$$

$$\leq \rho^{-\frac{s}{2}}\rho\kappa^2\|\widetilde{F}^{s-1}\|_2\sqrt{\|Y_0\|_{\mathcal{X}}^2 + \sum_{k=1}^\infty \rho^{-k+2}\|Y_k\|_{\mathcal{X}}^2}$$

$$\leq \rho^{-\frac{s}{2}}\rho\kappa^2\kappa^2\rho^{s-1}\sqrt{\|Y_0\|_{\mathcal{X}}^2 + \sum_{k=1}^\infty \rho^{-k+2}\|Y_k\|_{\mathcal{X}}^2}$$

$$= \kappa^4\rho^{\frac{s}{2}}\sqrt{\|Y_0\|_{\mathcal{X}}^2 + \sum_{k=1}^\infty \rho^{-k+2}\|Y_k\|_{\mathcal{X}}^2}.$$

Using $\rho^{-1+2} = \rho < 1$ for $k = 1$ in the above sum, the definition of $(\xi_k)$, and our assumption that $(Y_0, Y_1 \dots)$ has unit norm, we have

$$\|A^s((Y_0, Y_1, \dots))\| \leq \kappa^4 \rho^{\frac{s}{2}} \sqrt{\xi_0^2 \|Y_0\|_{\mathcal{X}}^2 + \sum_{k=1}^{\infty} \xi_k^2 \|Y_k\|_{\mathcal{X}}^2} = \kappa^4 \rho^{\frac{s}{2}}.$$

This completes the proof. ∎

**Lemma E.3.** *The* 2*-effective memory capacity is bounded above as*

$$H_2 \leq O\left(\kappa^4 (1 - \rho)^{-\frac{3}{2}}\right).$$

*Proof.* Using Lemma E.2 to bound $\|A^k\|$ for $k \geq 2$, we have

$$H_2 = \sqrt{\sum_{k=0}^{\infty} k^2 \|A^k\|^2} \leq O\left(\sqrt{\sum_{k=2}^{\infty} k^2 \kappa^8 \rho^k}\right) \leq O\left(\kappa^4 (1 - \rho)^{-\frac{3}{2}}\right). \qquad ∎$$

**Lemma E.4.** *Suppose* $R : \mathcal{X} \to \mathbb{R}$ *is defined by* $R(M) = \frac{1}{2} \|M\|_{\mathcal{X}}^2$. *Then, it is* 1*-strongly-convex and* $D = \max_{M, \widetilde{M} \in \mathcal{X}} |R(M) - R(\widetilde{M})| \leq d\kappa^8 (1 - \rho)^{-1}$.

*Proof.* Note that $R$ is 1-strongly-convex by definition. Using part 1 of Lemma E.1 and the definition of $\mathcal{X}$ (Eq. (10)), we have for all $M, \widetilde{M} \in \mathcal{X}$,

$$
\begin{aligned}
D &= \max_{M, \widetilde{M} \in \mathcal{X}} |R(M) - R(\widetilde{M})| \\
&= \max_{M, \widetilde{M} \in \mathcal{X}} \left| \frac{1}{2} \|M\|_{\mathcal{X}}^2 - \frac{1}{2} \|\widetilde{M}\|_{\mathcal{X}}^2 \right| \\
&\leq \max_{M \in \mathcal{X}} \|M\|_{\mathcal{X}}^2 \\
&= \max_{M \in \mathcal{X}} \sum_{s=1}^{\infty} \rho^{-s} \|M^{[s]}\|_F^2 && \text{by Eq. (11)} \\
&\leq \max_{M \in \mathcal{X}} \sum_{s=1}^{\infty} \rho^{-s} d \|M^{[s]}\|_2^2 && \text{by Lemma E.1} \\
&\leq \sum_{s=1}^{\infty} \rho^{-s} d\kappa^8 \rho^{2s} && \text{by Eq. (10)} \\
&\leq d\kappa^8 (1 - \rho)^{-1}.
\end{aligned}
$$

This completes the proof. ∎

**Lemma E.5.** *We can bound the norm of the state and control at time* $t$ *as*

$$\max\{\|s_t\|_2, \|u_t\|_2\} \leq D_{\mathcal{X}} = O\left(W \kappa^8 (1 - \rho)^{-2}\right).$$

*Proof.* We can bound the norm of $s_t$ and $u_t$ using Eqs. (14) and (15) as

$$
\begin{aligned}
\|s_t\|_2 &\leq \left\| \widetilde{F}^t s_0 + \sum_{k=0}^{t-1} \sum_{s=1}^{k+1} \widetilde{F}^{t-k-1} G M_k^{[s]} w_{k-s} + w_{t-1} \right\|_2 \\
&\leq \kappa^2 \rho^t + W + \sum_{k=0}^{t-1} \sum_{s=1}^{k+1} \kappa^2 \rho^{t-k-1} \kappa \kappa^4 \rho^s W \\
&\leq \kappa^2 + W + W \kappa^7 (1-\rho)^{-2} \\
&\leq O\left(W \kappa^7 (1-\rho)^{-2}\right). \\
\|u_t\|_2 &\leq \left\| K s_t + \sum_{s=1}^{t+1} M_t^{[s]} w_{t-s} \right\|_2 \\
&\leq O\left(W \kappa^8 (1-\rho)^{-2}\right) + \sum_{s=1}^{t+1} W \kappa^4 \rho^s \\
&\leq O\left(W \kappa^8 (1-\rho)^{-2}\right).
\end{aligned}
$$

Above, we used the assumptions that $\kappa, W \geq 1$. This completes the proof. ∎

**Lemma E.6.** *The Lipschitz constant of $f_t$ can be bounded above as*

$$
L \leq O\left(L_0 D_{\mathcal{X}} W \kappa (1-\rho)^{-1}\right),
$$

*where $D_{\mathcal{X}}$ is defined in Lemma E.5.*

*Proof.* Let $(M_0, \ldots, M_t)$ and $(\widetilde{M}_0, \ldots, \widetilde{M}_t)$ be two sequences of decisions, where $M_k$ and $\widetilde{M}_k \in \mathcal{X}$. Let $h_t$ and $\tilde{h}_t$ be the corresponding histories, and $(s_t, u_t)$ and $(\tilde{s}_t, \tilde{u}_t)$ be the corresponding state-control pairs at the end of round $t$. We have

$$
\begin{aligned}
\left| f_t(h_t) - f_t(\tilde{h}_t) \right| &= |c_t(s_t, u_t) - c_t(\tilde{s}_t, \tilde{u}_t)| \\
&\leq L_0 D_{\mathcal{X}} \max\{\|s_t - \tilde{s}_t\|_2, \|u_t - \tilde{u}_t\|_2\},
\end{aligned}
$$

where the last inequality follows from our assumptions about the functions $c_t$ and Lemma E.5. It suffices to bound the two norms on the right-hand side in terms of $\|h_t - \tilde{h}_t\|_{\mathcal{H}}$. For $k = 0, \ldots, t-1$,

define $Z_k^{[s]} = \widetilde{F}^{t-k-1} G(M_k^{[s]} - \widetilde{M}_k^{[s]})$. Using Eq. (14), we have

$$
\begin{aligned}
\|s_t - \tilde{s}_t\|_2 &= \left\| \sum_{k=0}^{t-1} \sum_{s=1}^{k+1} Z_k^{[s]} w_{k-s} \right\|_2 \\
&\leq \sum_{k=0}^{t-1} \sum_{s=1}^{k+1} \left\| Z_k^{[s]} w_{k-s} \right\|_2 \\
&= \sum_{k=0}^{t-1} \sum_{s=1}^{k+1} \left\| \rho^{-\frac{s}{2}} Z_k^{[s]} \rho^{\frac{s}{2}} w_{k-s} \right\|_2 \\
&\leq \sum_{k=0}^{t-1} \sum_{s=1}^{k+1} \left\| \rho^{-\frac{s}{2}} Z_k^{[s]} \right\|_2 \left\| \rho^{\frac{s}{2}} w_{k-s} \right\|_2 \\
&= \sum_{k=0}^{t-1} \sum_{s=1}^{k+1} \xi_{1+t-1-k} \left\| \rho^{-\frac{s}{2}} Z_k^{[s]} \right\|_2 \xi_{1+t-1-k}^{-1} \left\| \rho^{\frac{s}{2}} w_{k-s} \right\|_2 \\
&\leq \sqrt{\sum_{k=0}^{t-1} \sum_{s=1}^{k+1} \xi_{t-k}^2 \left\| \rho^{-\frac{s}{2}} Z_k^{[s]} \right\|_2^2} \sqrt{\sum_{k=0}^{t-1} \sum_{s=1}^{k+1} \xi_{t-k}^{-2} \left\| \rho^{\frac{s}{2}} w_{k-s} \right\|_2^2} \quad (16) \\
&= \underbrace{\sqrt{\sum_{k=0}^{t-1} \xi_{t-k}^2 \sum_{s=1}^{k+1} \left\| \rho^{-\frac{s}{2}} Z_k^{[s]} \right\|_2^2}}_{(a)} \underbrace{\sqrt{\sum_{k=0}^{t-1} \xi_{t-k}^{-2} \sum_{s=1}^{k+1} \left\| \rho^{\frac{s}{2}} w_{k-s} \right\|_2^2}}_{(b)},
\end{aligned}
$$

where Eq. (16) follows from the Cauchy-Schwarz inequality. The specific choice of weighted norms on $\mathcal{X}$ and $\mathcal{H}$ allow us to bound the terms (a) and (b) in terms of $\|h_t - \tilde{h}_t\|_{\mathcal{H}}$. We can bound the term (a) using the definition of $Z_k^{[s]}$, $\|\cdot\|_{\mathcal{X}}$, and $\|\cdot\|_{\mathcal{H}}$ as

$$
\begin{aligned}
\sqrt{\sum_{k=0}^{t-1} \xi_{t-k}^2 \sum_{s=1}^{k+1} \left\| \rho^{-\frac{s}{2}} Z_k^{[s]} \right\|_2^2} &= \sqrt{\sum_{k=0}^{t-1} \xi_{t-k}^2 \sum_{s=1}^{k+1} \rho^{-s} \left\| \widetilde{F}^{t-k-1} G(M_k^{[s]} - \widetilde{M}_k^{[s]}) \right\|_2^2} \\
&\leq \sqrt{\sum_{k=0}^{t-1} \xi_{t-k}^2 \sum_{s=1}^{k+1} \rho^{-s} \left\| \widetilde{F}^{t-k-1} G(M_k^{[s]} - \widetilde{M}_k^{[s]}) \right\|_F^2} \quad (17) \\
&\leq \|h_t - \tilde{h}_t\|_{\mathcal{H}}, \quad (18)
\end{aligned}
$$

where Eq. (17) follows from part 1 of Lemma E.1 and Eq. (18) follows from the definitions of $\|\cdot\|_{\mathcal{X}}$ and $\|\cdot\|_{\mathcal{H}}$. Using $\|w_t\|_2 \leq W$ for all rounds $t$, we can bound the term (b) as

$$
\begin{aligned}
\sqrt{\sum_{k=0}^{t-1} \xi_{t-k}^{-2} \sum_{s=1}^{k+1} \left\| \rho^{\frac{s}{2}} w_{k-s} \right\|_2^2} &\leq W \sqrt{\sum_{k=0}^{t-1} \xi_{t-k}^{-2} \sum_{s=1}^{k+1} \rho^s} \\
&\leq W \sqrt{\sum_{k=0}^{t-1} \xi_{t-k}^{-2} \frac{\rho(1 - \rho^{k+1})}{1 - \rho}} \\
&\leq W(1 - \rho)^{-1}, \quad (19)
\end{aligned}
$$

where Eq. (19) follows from the definition of $(\xi_k)$ (Eq. (13)). Substituting Eqs. (18) and (19) in Eq. (16), we have

$$
\|s_t - \tilde{s}_t\|_2 \leq W(1 - \rho)^{-1} \|h_t - \tilde{h}_t\|_{\mathcal{H}}.
$$

Similarly,

$$\|u_t - \tilde{u}_t\| = \left\| K(s_t - \tilde{s}_t) + \sum_{s=1}^{t+1}(M_t^{[s]} - \widetilde{M}_t^{[s]})w_{t-s} \right\|_2$$

$$\leq O\left( W\kappa(1-\rho)^{-1}\left\| h_t - \tilde{h}_t \right\|_{\mathcal{H}} \right),$$

where the last inequality follows from our assumption that $\|K\|_2 \leq \kappa$ and the above inequality for $\|s_t - \tilde{s}_t\|_2$. This completes the proof. ∎

**Lemma E.7.** *The Lipschitz constant of $\tilde{f}_t$ can be bounded above as*

$$\tilde{L} \leq O\left( L_0 D_\mathcal{X} W \kappa^5 (1-\rho)^{-\frac{3}{2}} \right),$$

*where $D_\mathcal{X}$ is defined in Lemma E.5.*

*Proof.* Using Lemma E.2 that bounds $\|A^k\|$, we have

$$\sqrt{\sum_{k=0}^{\infty} \|A^k\|^2} \leq O\left( \kappa^4(1-\rho)^{-\frac{1}{2}} \right).$$

Using Theorem 2.1 that bounds $\tilde{L}$ in terms of $L$ and the above, we have

$$\tilde{L} \leq O\left( L\kappa^4(1-\rho)^{-\frac{1}{2}} \right) \leq O\left( L_0 D_\mathcal{X} W \kappa^5(1-\rho)^{-\frac{3}{2}} \right),$$

where the last inequality follows from Lemma E.6. ∎

Now we restate and prove Theorem 4.1.

**Theorem 4.1.** *Consider the online linear control problem as defined in Section 4.1. Suppose the decisions in round $t$ are chosen using Algorithm 1. Then, the upper bound on the policy regret is*

$$O\left( L_0 W^2 \sqrt{T} d^{\frac{1}{2}} \kappa^{17}(1-\rho)^{-4.5} \right). \tag{2}$$

*Proof.* Using Theorem 3.1 and the above lemmas, we can upper bound the policy regret of Algorithm 1 for the online linear control problem by

$$O\left( \sqrt{\frac{D}{\alpha} T L \tilde{L} H_2} \right)$$

$$= O\left( \sqrt{d\kappa^8(1-\rho)^{-1} T \ (L_0 W^2 \kappa^9(1-\rho)^{-3})^2 \ \kappa^4(1-\rho)^{-\frac{1}{2}} \ \kappa^4(1-\rho)^{-\frac{3}{2}}} \right)$$

$$= O\left( L_0 W^2 \sqrt{T} d^{\frac{1}{2}} \kappa^{17}(1-\rho)^{-4.5} \right).$$

This completes the proof. ∎

### E.3 Existing Regret Bound

The upper bound on policy regret for the online linear control problem in existing work is given in Agarwal et al. [2019b, Theorem 5.1]. The theorem statement only shows the dependence on $\tilde{L}, W$, and $T$. The dependence on $d, \kappa$, and $\rho$ can be found in the details of the proof. Below we give a detailed accounting of all of these terms in their regret bound.

To simplify notation let $\gamma = 1 - \rho$. Agarwal et al. [2019b] define

$$H = \frac{\kappa^2}{\gamma}\log(T) \quad \text{and} \quad C = \frac{W(\kappa^2 + H\kappa_B\kappa^2 a)}{\gamma(1 - \kappa^2(1-\gamma)^{H+1})} + \frac{\kappa_B\kappa^3 W}{\gamma}.$$

The value of $a$ is not specified in Theorem 5.1. However, from Theorem 5.3 and the definition of $\mathcal{M}$ in Algorithm 1 their paper, we can infer that $a = \kappa_B\kappa^3$.

The final regret bound is obtained by summing Equations 5.1, 5.3, and 5.4. Given the definition of $H$ above, we have that

$$(1-\gamma)^{H+1} \leq \exp(-\kappa^2 \log T) = T^{-\kappa^2}.$$

So, the dominant term in the regret bound is Equation 5.4, which is

$$O\left(L_0 W C d^{\frac{3}{2}} \kappa_B^2 \kappa^6 H^{2.5} \gamma^{-1} \sqrt{T}\right).$$

Substituting the values of $H$ and $C$ from above and collecting terms, we have that the upper bound on policy regret in existing work [Agarwal et al., 2019b, Theorem 5.1] is

$$
\begin{aligned}
&O\left(L_0 W d^{\frac{3}{2}} \sqrt{T} \log(T)^{2.5} \kappa_B^2 \kappa^{11} \gamma^{-3.5} C\right) \\
&= O\left(L_0 W d^{\frac{3}{2}} \sqrt{T} \log(T)^{2.5} \kappa_B^2 \kappa^{11} \gamma^{-3.5} \left(\frac{W(\kappa^2 + H\kappa_B\kappa^2 a)}{\gamma(1 - \kappa^2(1-\gamma)^{H+1})} + \frac{\kappa_B \kappa^3 W}{\gamma}\right)\right) \\
&= O\left(L_0 W d^{\frac{3}{2}} \sqrt{T} \log(T)^{2.5} \kappa_B^2 \kappa^{11} \gamma^{-3.5} \left(\frac{W\kappa^2}{\gamma(1 - \kappa^2(1-\gamma)^{H+1})} + \frac{W\kappa_B^2 \kappa^7 \log(T)}{\gamma^2(1 - \kappa^2(1-\gamma)^{H+1})} + \frac{\kappa_B \kappa^3 W}{\gamma}\right)\right) \\
&= O\left(L_0 W^2 d^{\frac{3}{2}} \sqrt{T} \log(T)^{2.5} \kappa_B^2 \kappa^{13} \gamma^{-4.5}(1 - \kappa^2(1-\gamma)^{H+1})^{-1}\right) \\
&\quad + O\left(L_0 W^2 d^{\frac{3}{2}} \sqrt{T} \log(T)^{3.5} \kappa_B^4 \kappa^{18} \gamma^{-5.5}(1 - \kappa^2(1-\gamma)^{H+1})^{-1}\right) \\
&\quad + O\left(L_0 W^2 d^{\frac{3}{2}} \sqrt{T} \log(T)^{2.5} \kappa_B^3 \kappa^{14} \gamma^{-4.5}\right) \\
&= O\left(L_0 W^2 d^{\frac{3}{2}} \sqrt{T} \log(T)^{3.5} \kappa_B^4 \kappa^{18} \gamma^{-5.5}\right).
\end{aligned}
$$

Above we used that $\lim_{T\to\infty}(1 - \kappa^2(1-\gamma)^{H+1})^{-1} = 1$ to simplify the expressions. Therefore, the upper bound on policy regret for the online linear control problem in existing work is

$$O\left(L_0 W^2 d^{\frac{3}{2}} \sqrt{T} \log(T)^{3.5} \kappa_B^4 \kappa^{18} \gamma^{-5.5}\right). \tag{20}$$

# F   Online Performative Prediction

Before formulating the online performative prediction problem in our OCO with unbounded memory framework, we state the definition of 1-Wasserstein distance that we use in our regret analysis. Informally, the 1-Wasserstein distance is a measure of the distance between two probability measures.

**Definition F.1** (1-Wasserstein Distance). Let $(\mathcal{Z}, d)$ be a metric space. Let $\mathbb{P}(\mathcal{Z})$ denote the set of Radon probability measures $\nu$ on $\mathcal{Z}$ with finite first moment. That is, there exists $z' \in \mathcal{Z}$ such that $\mathbb{E}_{z\sim\nu}[d(z, z')] < \infty$. The 1-Wasserstein distance between two probability measures $\nu, \nu' \in \mathbb{P}(\mathcal{Z})$ is defined as

$$W_1(\nu, \nu') = \sup\{\mathbb{E}_{z\sim\nu}[f(z)] - \mathbb{E}_{z\sim\nu'}[f(z)]\},$$

where the supremum is taken over all 1-Lipschitz continuous functions $f : \mathcal{Z} \to \mathbb{R}$.

## F.1   Formulation as OCO with Unbounded Memory

Now we formulate the online performative prediction problem in our framework by defining the decision space $\mathcal{X}$, the history space $\mathcal{H}$, and the linear operators $A : \mathcal{H} \to \mathcal{H}$ and $B : \mathcal{W} \to \mathcal{H}$. Then, we define the functions $f_t : \mathcal{H} \to \mathbb{R}$ in terms of $l_t$ and finally, prove an upper bound on the policy regret. For notational convenience, let $(y_k)$ denote the sequence $(y_0, y_1, \dots)$.

Let $\rho \in (0, 1)$. Let the decision space $\mathcal{X} \subseteq \mathbb{R}^d$ be closed and convex with $\|\cdot\|_{\mathcal{X}} = \|\cdot\|_2$. Let the history space $\mathcal{H}$ be the $\ell^1$-direct sum of countably infinte number of copies of $\mathcal{X}$. Define the linear operators $A : \mathcal{H} \to \mathcal{H}$ and $B : \mathcal{X} \to \mathcal{H}$ as

$$A((y_0, y_1, \dots)) = (0, \rho y_0, \rho y_1, \dots) \quad \text{and} \quad B(x) = (x, 0, \dots).$$

Note that the problem is an OCO with $\rho$-discounted infinite memory problem and follows linear sequence dynamics with the 1-norm (Definition 2.3).

Given a sequence of decisions $(x_k)_{k=1}^t$, the history is $h_t = (x_t, \rho x_{t-1}, \dots, \rho^{t-1} x_1, 0, \dots)$ and the data distribution $p_t = p_t(h_t)$ satisfies:

$$z \sim p_t \text{ iff } z \sim \sum_{k=1}^{t-1} (1 - \rho)\rho^{k-1}(\xi + Fx_{t-k}) + \rho^t p_1. \tag{21}$$

This follows from the recursive definition of $p_t$ and parametric assumption about $\mathcal{D}(x)$. Define the functions $f_t : \mathcal{H} \to [0, 1]$ by

$$f_t(h_t) = \mathbb{E}_{z \sim p_t}[l_t(x_t, z)].$$

With the above formulation and definition of $f_t$, the original goal of minimizing the difference between the algorithm's total loss and the total loss of the best fixed decision is equivalent to minimizing the policy regret,

$$\sum_{t=1}^T f_t(h_t) - \min_{x \in \mathcal{X}} \sum_{t=1}^T \tilde{f}_t(x).$$

### F.2 Regret Analysis

**Lemma F.1.** *The operator norm $\|A^s\|$ is bounded above as*

$$\|A^s\| \leq O\left(\rho^s\right).$$

*Proof.* Recall the definition of $\mathcal{H}$ and $\|\cdot\|_{\mathcal{H}}$. Let

$$(y_0, y_1, \dots) = (x_0, \rho x_1, \rho^2 x_2, \dots)$$

be an element of $\mathcal{H}$ with unit norm, i.e.,

$$\sum_{k=0}^{\infty} \|y_k\| = 1.$$

From the definition of the operator $A$, we have

$$A^s((y_0, y_1, \dots)) = (0, \dots, 0, \rho^s x_0, \rho^{s+1} x_1, \dots).$$

Now we bound $\|A^s\|$ as follows. By definition of $A^s$ and $\|\cdot\|_{\mathcal{H}}$, we have

$$\|A^s((y_0, y_1, \dots))\| = \sum_{k=0}^{\infty} \rho^{s+k} \|x_k\| = \rho^s \sum_{k=0}^{\infty} \rho^k \|x_k\| = \rho^s \sum_{k=0}^{\infty} \|y_k\| = \rho^s. \qquad \blacksquare$$

**Lemma F.2.** *The $1$-effective memory capacity is bounded above as*

$$H_2 \leq O\left((1 - \rho)^{-2}\right).$$

*Proof.* Using Lemma F.1 to bound $\|A^k\|$, we have

$$H_1 = \sum_{k=0}^{\infty} k\|A^k\| = \sum_{k=0}^{\infty} k\rho^k \leq O\left((1 - \rho)^{-2}\right). \qquad \blacksquare$$

**Lemma F.3.** *Suppose $R : \mathcal{X} \to \mathbb{R}$ is defined by $R(x) = \frac{1}{2}\|x\|_{\mathcal{X}}^2$. Then, it is $1$-strongly-convex and $D = \max_{x, \tilde{x} \in \mathcal{X}} |R(x) - R(\tilde{x})| \leq D_{\mathcal{X}}^2$.*

*Proof.* Note that $R$ is $1$-strongly-convex by definition. By the assumption that $\|x\|_{\mathcal{X}} \leq D_{\mathcal{X}}$ for all $x \in \mathcal{X}$, we have that $D \leq D_{\mathcal{X}}^2$. $\qquad \blacksquare$

**Lemma F.4.** *The Lipschitz constant of $f_t$ can be bounded above as*

$$L \leq O\left(L_0 \frac{1 - \rho}{\rho} \|F\|_2\right).$$

*Proof.* Let $(x_1, \ldots, x_t)$ and $(\tilde{x}_1, \ldots, \tilde{x}_t)$ be two sequences of decisions, where $x_k, \tilde{x}_k \in \mathcal{X}$. Let $h_t$ and $\tilde{h}_t$ be the corresponding histories, and $p_t$ and $\tilde{p}_t$ be the corresponding distributions at the end of round $t$. We have

$$
\left| f_t(h_t) - f_t(\tilde{h}_t) \right|
$$
$$
= |\mathbb{E}_{z \sim p_t} [l_t(x_t, z)] - \mathbb{E}_{z \sim \tilde{p}_t} [l_t(\tilde{x}_t, z)]|
$$
$$
= |\mathbb{E}_{z \sim p_t} [l_t(x_t, z)] - \mathbb{E}_{z \sim p_t} [l_t(\tilde{x}_t, z)] + \mathbb{E}_{z \sim p_t} [l_t(\tilde{x}_t, z)] - \mathbb{E}_{z \sim \tilde{p}_t} [l_t(\tilde{x}_t, z)]|
$$
$$
\leq L_0 \|x_t - \tilde{x}_t\|_2 + L_0 W_1(p_t, \tilde{p}_t),
$$

where the last inequality follows from the assumptions about the functions $l_t$ and the definition of the Wasserstein distance $W_1$. By definition of $p_t$ (Eq. (21)), we have

$$
W_1(p_t, \tilde{p}_t) \leq \sum_{k=1}^{t-1} \frac{1-\rho}{\rho} \rho^k \|F\|_2 \|x_{t-k} - \tilde{x}_{t-k}\|_2
$$
$$
\leq \frac{1-\rho}{\rho} \|F\|_2 \|h_t - \tilde{h}_t\|_{\mathcal{H}},
$$

where the last inequality follows from the definition of $\| \cdot \|_{\mathcal{H}}$. Therefore, $L \leq L_0 \frac{1-\rho}{\rho} \|F\|_2$. ∎

**Lemma F.5.** *The Lipschitz constant of $f_t$ can be bounded above as*

$$
\tilde{L} \leq O \left( L_0 \frac{1}{\rho} \|F\|_2 \right).
$$

*Proof.* Using Lemma F.1 that bounds $\|A^k\|$, we have

$$
\sum_{k=0}^{\infty} \|A^k\| = (1-\rho)^{-1}.
$$

Using Theorem 2.1 that bounds $\tilde{L}$ in terms of $L$ and the above, we have

$$
\tilde{L} \leq O \left( L(1-\rho)^{-1} \right) = O \left( L_0 \frac{1}{\rho} \|F\|_2 \right),
$$

where the last equality follows from Lemma F.4. ∎

Now we restate and prove Theorem 4.2.

**Theorem 4.2.** *Consider the online performative prediction problem as defined in Section 4.2. Suppose the decisions in round $t$ are chosen using Algorithm 1. Then, the upper bound on the policy regret is*

$$
O \left( D_{\mathcal{X}} L_0 \sqrt{T} \|F\|_2 (1-\rho)^{-\frac{1}{2}} \rho^{-1} \right).
$$

*Proof.* Using Theorem 3.1 and the above lemmas, we can upper bound the policy regret of Algorithm 1 for the online performative prediction problem by

$$
O \left( \sqrt{\frac{D}{\alpha} T L \tilde{L} H_1} \right) = O \left( D_{\mathcal{X}} L_0 \|F\|_2 (1-\rho)^{-\frac{1}{2}} \rho^{-1} \sqrt{T} \right).
$$

This completes the proof. ∎

We note that the upper bound can be improved by defining a weighted norm on $\mathcal{H}$ similar to the approach in Appendix E. However, here we present the looser anaysis for simplicity of exposition.

# G   Implementation Details for Algorithm 1

In this section we discuss how to implement Algorithm 1 efficiently.

**Dimensionality of $\mathcal{X}$.** First, note that the decisions $x \in \mathcal{X}$ could be high-dimensional, e.g., an unbounded sequence of matrices as in the online linear control problem, but this is external to our framework and is application dependent. Our framework can be applied to $\mathcal{X}$ or to a lower-dimensional decision space $\mathcal{X}'$. However, the choice of $\mathcal{X}'$ and analyzing the difference

$$\min_{x' \in \mathcal{X}'} \sum_{t=1}^{T} \tilde{f}_t(x') - \min_{x \in \mathcal{X}} \sum_{t=1}^{T} \tilde{f}_t(x)$$

is application dependent. For example, for the online linear control problem one could consider a restricted class of disturbance-action controllers that operate on a constant number of past disturbances as opposed to all the past disturbances, and then analyze the difference between these two policy classes. See, for example, Agarwal et al. [2019b, Lemma 5.2].

**Computational cost of each iteration of Algorithm 1.** Now we discuss how to implement each iteration of Algorithm 1 efficiently. We are interested in the computational cost of computing the decision $x_{t+1}$ as a function of $t$. (Given the above discussion about the dimensionality of $\mathcal{X}$, we ignore the fact that the dimensionality of the decisions themselves could depend on $t$.) Therefore, for the purposes of this section we (i) use $O(\cdot)$ notation to hide absolute constants and problem parameters excluding $t$ and $T$; (ii) invoke the operators $A$ and $B$ by calling oracles $\mathcal{O}_A(\cdot)$ and $\mathcal{O}_B(\cdot)$; and (iii) evaluate the functions $f_t$ by calling oracles $\mathcal{O}_f(t, \cdot)$. Recall from Assumption **A1** that we assume the learner knows the operators $A$ and $B$, and observes $f_t$ at the end of each round $t$. So, the oracles $\mathcal{O}_A, \mathcal{O}_B$, and $\mathcal{O}_f$ are readily available.

Algorithm 1 chooses the decision $x_{t+1}$ as

$$x_{t+1} \in \arg\min_{x \in \mathcal{X}} \sum_{s=1}^{t} \tilde{f}_s(x) + \frac{R(x)}{\eta} = \arg\min_{x \in \mathcal{X}} \underbrace{\sum_{s=1}^{t} f_s\left(\sum_{k=0}^{s-1} A^k B x\right) + \frac{R(x)}{\eta}}_{=F_t(x)}.$$

Since $F_t(x)$ is a sum of $f_1, \ldots, f_t$, evaluating $F_t(x)$ requires $\Theta(t)$ oracle calls to $\mathcal{O}_f$. However, this issue is present in FTRL for OCO and OCO with finite memory as well and is not specific to our framework. To deal with this issue, one could consider mini-batching algorithms [Dekel et al., 2012, Altschuler and Talwar, 2018, Chen et al., 2020] such as Algorithm 2.

A naïve implementation to evaluate $F_t(x)$ could require $O(t^3)$ oracle calls to $\mathcal{O}_A$: for each $s \in [t]$, constructing the argument $\sum_{k=0}^{s-1} A^k B x$ for $f_s$ could require $k$ oracle calls to $\mathcal{O}_A$ to compute $A^k B x$, for a total of $O(s^2)$ oracle calls. However, $F_t(x)$ can be evaluated with just $O(t)$ oracle calls to $\mathcal{O}_A$ by constructing the arguments incrementally. For $t \geq 0$, define $\Gamma_t : \mathcal{X} \to \mathcal{H}$ as

$$\Gamma_0(x) = Bx$$
$$\Gamma_t(x) = A\left(\Gamma_{t-1}(x)\right) \quad \text{for } t \geq 1.$$

Note that $\Gamma_t(Bx) = A^t B x$. Also, for $t \geq 1$, define $\Phi_t : \mathcal{X} \to \mathcal{H}$ as

$$\Phi_1(x) = \Gamma_0(x)$$
$$\Phi_t(x) = \Phi_{t-1}(x) + \Gamma_{t-1}(x) \quad \text{for } t \geq 2.$$

Note that $\Phi_s(x) = \sum_{k=0}^{s-1} A^k B x$ is the argument for $f_s$. These can be constructed incrementally as follows.

1. Construct $\Gamma_0(x)$ using one oracle call to $\mathcal{O}_B$.
2. For $s = 1$,
   (a) Construct $\Phi_1(x) = \Gamma_0(x)$.
   (b) Construct $\Gamma_1(x)$ from $\Gamma_0(x)$ using one oracle call to $\mathcal{O}_A$.
3. For $s \geq 2$,
   (a) Construct $\Phi_s(x)$ by adding $\Phi_{s-1}(x)$ and $\Gamma_{s-1}(x)$. This can be done in $O(1)$ time. Recall from our earlier discussion that $O(\cdot)$ hides absolute constants and problem parameters excluding $t$ and $T$.
   (b) Construct $\Gamma_s(x)$ from $\Gamma_{s-1}(x)$ using one oracle call to $\mathcal{O}_A$.

By incrementally constructing $\Phi_s(x)$ as above, we can evaluate $F_t(x)$ in $O(t)$ time with $O(1)$ oracle calls to $\mathcal{O}_B$, $O(t)$ oracle calls to $\mathcal{O}_A$, and $O(t)$ oracle calls to $\mathcal{O}_f$.

**Memory usage of Algorithm 1.**   We end with a brief discussion of the memory usage of Algorithm 1. We are interested in the memory usage of computing the decision $x_{t+1}$ as a function of $t$. (Given the discussion about the dimensionality of $\mathcal{X}$ at the start of this section, we ignore the fact that the dimensionality of the decisions themselves could depend on $t$.) For each $t \in [T]$, the memory usage could be as low as $O(1)$ (if, for example, $\mathcal{X} \subseteq \mathbb{R}^d$, and $A, B \in \mathbb{R}^{d \times d}$, which implies that $\Phi_t(x)$ is a $d$-dimensional vector) or as high as $O(t)$ (if, for example, $\Phi_t(x)$ is a $t$-length sequence of $d$-dimensional vectors). However, the memory usage is already $\Omega(t)$ to store the functions $f_1, \ldots, f_t$. Therefore, Algorithm 1 only incurs a constant factor overhead.

# H   An Algorithm with A Low Number of Switches: Mini-Batch FTRL

In this section we present an algorithm (Algorithm 2) for OCO with unbounded memory that provides the same upper bound on policy regret as Algorithm 1 while guaranteeing a small number of switches. Algorithm 2 combines FTRL on the functions $\tilde{f}_t$ with a mini-batching approach. First, it divides rounds into batches of size $S$, where $S$ is a parameter. Second, at the start of batch $b \in \{1, \ldots, \lceil T/S \rceil\}$, it performs FTRL on the functions $\{g_1, \ldots, g_b\}$, where $g_i$ is the average of the functions $\tilde{f}_t$ in batch $i$. Then, it uses this decision for the entirety of the current batch. By design, Algorithm 2 switches decisions at most $O(T/S)$ times. This algorithm is insipired by similar algorithms for online learning and OCO [Dekel et al., 2012, Altschuler and Talwar, 2018, Chen et al., 2020].

---

**Algorithm 2:** `Mini-Batch FTRL`

**Input** : Time horizon $T$, step size $\eta$, $\alpha$-strongly-convex regularizer $R : \mathcal{X} \to \mathbb{R}$, batch size $S$.

1 Initialize history $h_0 = 0$.
2 **for** $t = 1, 2, \ldots, T$ **do**
3  | **if** $t \mod S = 1$ **then**
4  |  | Let $N_t = \{1, \ldots, \lceil \frac{t}{S} \rceil\}$ denote the number of batches so far.
5  |  | For $b \in N_t$, let $T_b = \{(b-1)S + 1, \ldots, bS\}$ denote the rounds in batch $b$.
6  |  | For $b \in N_t$, let $g_b = \frac{1}{S} \sum_{s \in T_b} \tilde{f}_s$. denote the average of the functions in batch $b$.
7  |  | Learner chooses $x_t \in \arg\min_{x \in \mathcal{X}} \sum_{b \in N_t} g_b(x) + \frac{R(x)}{\eta}$.
8  | **end**
9  | **else**
10 |  | Learner chooses $x_t = x_{t-1}$.
11 | **end**
12 | Set $h_t = A h_{t-1} + B x_t$.
13 | Learner suffers loss $f_t(h_t)$ and observes $f_t$.
14 **end**

---

**Theorem H.1.** *Consider an online convex optimization with unbounded memory problem specified by* $(\mathcal{X}, \mathcal{H}, A, B)$. *Let the regularizer* $R : \mathcal{X} \to \mathbb{R}$ *be* $\alpha$-*strongly-convex and satisfy* $|R(x) - R(\tilde{x})| \le D$ *for all* $x, \tilde{x} \in \mathcal{X}$. *Algorithm 2 with batch size* $S$ *and step-size* $\eta$ *satisfies*

$$R_T(\texttt{Mini-Batch FTRL}) \le \frac{SD}{\eta} + \eta \frac{T\tilde{L}^2}{\alpha} + \eta \frac{TL\tilde{L}H_1}{S\alpha}.$$

*If* $\eta = \sqrt{\frac{\alpha SD}{T\tilde{L}\left(\frac{LH_1}{S} + \tilde{L}\right)}}$, *then*

$$R_T(\texttt{Mini-Batch FTRL}) \le O\left(\sqrt{\frac{D}{\alpha} T \left(L\tilde{L}H_1 + S\tilde{L}^2\right)}\right).$$

Setting the batch size to be $S = LH_1/\tilde{L}$ we obtain the same upper bound on policy regret as Algorithm 1 while guaranteeing that the decisions $x_t$ switch at most $T\tilde{L}/LH_1$ times.

**Corollary H.1.** *Consider an online convex optimization with unbounded memory problem specified by* $(\mathcal{X}, \mathcal{H}, A, B)$. *Let the regularizer* $R : \mathcal{X} \to \mathbb{R}$ *be* $\alpha$-*strongly-convex and satisfy* $|R(x) - R(\tilde{x})| \le D$

*for all $x, \tilde{x} \in \mathcal{X}$. Algorithm 2 with batch size $S = \frac{LH_1}{\tilde{L}}$ and step-size $\eta = \sqrt{\frac{\alpha SD}{T\tilde{L}\left(\frac{LH_1}{S}+\tilde{L}\right)}}$ satisfies*

$$R_T(\texttt{Mini-Batch FTRL}) \leq O\left(\sqrt{\frac{D}{\alpha}TL\tilde{L}H_1}\right).$$

*Furthermore, the decisions $x_t$ switch at most $\frac{T\tilde{L}}{LH_1}$ times.*

Intuitively, in the OCO with unbounded memory framework each decision $x_t$ is penalized not just in round $t$ but in future rounds as well. Therefore, instead of immediately changing the decision, it is prudent to stick to it for a while, collect more data, and then switch decisions. For the OCO with finite memory problem, the constant memory length $m$ provides a natural measure of how long decisions penalized for and when one should switch decisions. In the general case, this is measured by the quantity $LH_1/\tilde{L}$. Note that this simplifies to $m$ for OCO with finite memory for all $p$-norms.

*Proof of Theorem H.1.* For simplicity, assume that $T$ is a multiple of $S$. Otherwise, the same proof works after replacing $\frac{T}{S}$ with $\lceil\frac{T}{S}\rceil$. Let $x^* \in \arg\min_{x\in\mathcal{X}}\sum_{t=1}^T \tilde{f}_t(x)$. Note that we can write the regret as

$$R_T(\texttt{Mini-Batch FTRL}) = \sum_{t=1}^T f_t(h_t) - \min_{x\in\mathcal{X}}\sum_{t=1}^T \tilde{f}_t(x)$$

$$= \underbrace{\sum_{t=1}^T f_t(h_t) - \tilde{f}_t(x_t)}_{(a)} + \underbrace{\sum_{t=1}^T \tilde{f}_t(x_t) - \tilde{f}_t(x^*)}_{(b)}.$$

We can bound the term (b) using Theorem B.1 for mini-batches [Dekel et al., 2012, Altschuler and Talwar, 2018, Chen et al., 2020] by

$$\frac{SD}{\eta} + \eta\frac{T\tilde{L}^2}{\alpha}.$$

It remains to bound term (a). Let $N = T/S$ denote the number of batches and $T_n = \{(n-1)S + 1, \ldots, nS\}$ denote the rounds in batch $n \in [N]$. We can write

$$\sum_{t=1}^T f_t(h_t) - \tilde{f}_t(x_t) = \sum_{t=1}^T f_t\left(\sum_{k=0}^{t-1}A^kBx_{t-k}\right) - f_t\left(\sum_{k=0}^{t-1}A^kBx_t\right) \qquad \text{by Definition 2.1}$$

$$\leq L\sum_{t=1}^T \left\|\sum_{k=0}^{t-1}A^kBx_{t-k} - \sum_{k=0}^{t-1}A^kBx_t\right\| \qquad \text{by Assumption A4}$$

$$\leq \frac{T}{S}L\underbrace{\sum_{t\in T_N}\left\|\sum_{k=0}^{t-1}A^kBx_{t-k} - \sum_{k=0}^{t-1}A^kBx_t\right\|}_{(c)},$$

where the last inequality follows because of the following. Consider rounds $t_1 = b_1S + r$ and $t_2 = b_2S + r$ for $b_1 < b_2$ and $r \in [S]$. Then, $\|h_{t_1} - \sum_{k=0}^{t_1-1}A^kBx_{t_1}\| \leq \|h_{t_2} - \sum_{k=0}^{t_2-1}A^kBx_{t_2}\|$ because the latter sums over more terms in its history and decisions in consecutive batches have distance bounded above by $\eta\tilde{L}/\alpha$ (Theorem B.1). Therefore, it suffices to show that term (c) is upper bounded by $\eta\tilde{L}H_1/\alpha$. We have

$$\sum_{t\in T_N}\left\|\sum_{k=0}^{t-1}A^kBx_{t-k} - \sum_{k=0}^{t-1}A^kBx_t\right\| \leq \sum_{t\in T_N}\sum_{k=0}^{t-1}\left\|A^kBx_{t-k} - A^kBx_t\right\|$$

$$\leq \sum_{t\in T_N}\sum_{k=0}^{t-1}\|A^k\|\|B\|\|x_{t-k} - x_t\|$$

$$\leq \sum_{t\in T_N}\sum_{k=0}^{t-1}\|A^k\|\|x_{t-k} - x_t\| \qquad \text{by Assumption A2.}$$

Since the same decision $x_n$ is chosen in all rounds of batch $n$, we can reindex and rewrite

$$\sum_{t \in T_N} \left\| \sum_{k=0}^{t-1} A^k B x_{t-k} - \sum_{k=0}^{t-1} A^k B x_t \right\| \leq \sum_{t \in T_N} \sum_{k=0}^{t-1} \|A^k\| \|x_{t-k} - x_t\|$$

$$\leq \sum_{o=0}^{S-1} \sum_{n=1}^{N-1} \sum_{s=1}^{S} \|A^{(N-n-1)S+s+o}\| \|x_N - x_n\|$$

$$\leq \eta \frac{\tilde{L}}{\alpha} \sum_{o=0}^{S-1} \sum_{n=1}^{N-1} \sum_{s=1}^{S} (N-n) \|A^{(N-n-1)S+s+o}\|$$

$$= \eta \frac{\tilde{L}}{\alpha} \sum_{o=0}^{S-1} \sum_{n=1}^{N-1} \sum_{s=1}^{S} n \|A^{(n-1)S+s+o}\|,$$

where the last inequality follows from bounding the distance between decision in consecutive batches Theorem B.1 and the triangle inequality. Expanding the triple sum yields

$$\sum_{o=0}^{S-1} \sum_{n=1}^{N-1} \sum_{s=1}^{S} n \|A^{(n-1)S+s+o}\|$$

$$\leq \|A\| + \cdots + \|A^S\| + 2\|A^{S+1}\| + \cdots + 2\|A^{2S}\| + 3\|A^{2S+1}\| + \cdots + 3\|A^{3S}\| + \ldots$$

$$+ \|A^2\| + \cdots + \|A^{S+1}\| + 2\|A^{S+2}\| + \cdots + 2\|A^{2S+1}\| + 3\|A^{2S+2}\| + \cdots + 3\|A^{3S+1}\| + \ldots$$

$$\vdots$$

$$+ \|A^S\| + \cdots + \|A^{2S-1}\| + 2\|A^{2S}\| + \cdots + 2\|A^{3S-1}\| + 3\|A^{3S}\| + \cdots + 3\|A^{4S-1}\| + \ldots,$$

where each line above corresponds to a value of $o \in \{0, \ldots, S-1\}$. Adding up these terms yields $H_1$. This completes the proof. ∎

Note that Theorem H.1 only provides an upper bound on the policy regret for the general case. Unlike Algorithm 1, it is unclear how to obtain a stronger bound depending on $H_p$ for the case of linear sequence dynamics with the $\xi$-weighted $p$-norm for $p > 1$. The above proof can be specialized for this special case, similar to the proofs of Theorem 2.1 and Lemma C.1, to obtain

$$\sum_{t \in T_N} \left\| \sum_{k=0}^{t-1} A^k B x_{t-k} - \sum_{k=0}^{t-1} A^k B x_t \right\| \leq \eta \frac{\tilde{L}}{\alpha} \sum_{o=0}^{S-1} \left( \sum_{n=1}^{N-1} \sum_{s=1}^{S} \left( n \|A^{(n-1)S+s+o}\| \right)^p \right)^{\frac{1}{p}}$$

and

$$\sum_{o=0}^{S-1} \left( \sum_{n=1}^{N-1} \sum_{s=1}^{S} \left( n \|A^{(n-1)S+s+o}\| \right)^p \right)^{\frac{1}{p}}$$

$$\leq \left( \|A\|^p + \cdots + \|A^S\|^p + 2^p \|A^{S+1}\|^p + \ldots 2^p \|A^{2S}\|^p + 3^p \|A^{2S+1}\|^p + \ldots \right)^{\frac{1}{p}}$$

$$+ \left( \|A^2\|^p + \cdots + \|A^{S+1}\|^p + 2^p \|A^{S+2}\|^p + \ldots 2^p \|A^{2S+1}\|^p + 3^p \|A^{2S+2}\|^p + \ldots \right)^{\frac{1}{p}}$$

$$\vdots$$

$$\left( \|A^S\|^p + \cdots + \|A^{2S-1}\|^p + 2^p \|A^{2S}\|^p + \ldots 2^p \|A^{3S-1}\|^p + 3^p \|A^{3S}\|^p + \ldots \right)^{\frac{1}{p}}.$$

The above expression cannot be easily simplified to $O(H_p)$. However, for the special case of OCO with finite memory, which follows linear sequence dynamics with the 2-norm, we can do so by leveraging the special structure of the linear operator $A_{\text{finite},m}$.

**Theorem H.2.** *Consider an online convex optimization with finite memory problem with constant memory length $m$ specified by $(\mathcal{X}, \mathcal{H} = \mathcal{X}^m, A_{\text{finite},m}, B_{\text{finite},m})$. Let the regularizer $R : \mathcal{X} \to \mathbb{R}$ be $\alpha$-strongly-convex and satisfy $|R(x) - R(\tilde{x})| \leq D$ for all $x, \tilde{x} \in \mathcal{X}$. Algorithm 2 with batch size $m$*

*and step-size $\eta = \sqrt{\frac{\alpha m D}{T\tilde{L}\left(Lm^{\frac{1}{2}}+\tilde{L}\right)}}$ satisfies*

$$R_T(\texttt{Mini-Batch FTRL}) \leq O\left(\sqrt{\frac{D}{\alpha}TL\tilde{L}m^{\frac{3}{2}}}\right) \leq O\left(m\sqrt{\frac{D}{\alpha}TL^2}\right).$$

*Furthermore, the decisions $x_t$ switch at most $\frac{T}{m}$ times.*

*Proof.* Given the proof of Theorem H.1 and the above discussion, it suffices to show that

$$\sum_{o=0}^{S-1}\left(\sum_{n=1}^{N-1}\sum_{s=1}^{S}\left(n\|A^{(n-1)S+s+o}\|\right)^2\right)^{\frac{1}{2}} \leq H_2 = m^{\frac{3}{2}}.$$

Recall that $\|A_{\text{finite}}^k\| = 1$ if $k \leq m$ and 0 otherwise. Using this and $S = m$, we have that the above sum is at most $\sqrt{m} + \sqrt{m-1} + \cdots + \sqrt{1} = O\left(m^{\frac{3}{2}}\right)$. This completes the proof. ∎

## I  Experiments

In this section we present some simple simulation experiments.[2]

**Problem Setup.** We consider the problem of online linear control with a constant input controller class $\Pi = \{\pi_u : \pi(s) = u \in \mathcal{U}\}$. Let $T$ denote the time horizon. Let $\mathcal{S} = \mathbb{R}^d$ and $\mathcal{U} = \{u \in \mathbb{R}^d : \|u\|_2 \leq 1\}$ denote the state and control spaces. Let $s_t$ and $u_t$ denote the state and control at time $t$ with $s_0$ being the initial state. The system evolves according to linear dynamics $s_{t+1} = Fs_t + Gu_t + w_t$, where $F, G \in \mathbb{R}^{d\times d}$ are system matrices and $w_t \in \mathbb{R}^d$ is a disturbance. The loss function in round $t$ is simply $c_t(s_t, u_t) = c_t(s_t) = \sum_{j=1}^d s_{t,j}$, where $s_{t,j}$ denotes the $j$-th coordinate of $s_t$. The goal is to choose a sequence of control inputs $u_0, \ldots, u_{T-1} \in \mathcal{U}$ to minimize the regret

$$\sum_{t=0}^{T-1} c_t(s_t, u_t) - \min_{u \in \mathcal{U}} \sum_{t=0}^{T-1} c_t(s_t^u, u),$$

where $s_t^u$ denotes the state in round $t$ upon choosing control input $u$ in each round. Note that the state in round $t$ can be written as

$$s_t = \sum_{k=1}^t F^k Gu_{t-k} + \sum_{k=1}^t F^k w_{t-k}.$$

Therefore, we can formulate this problem as an OCO with unbounded memory problem by setting $\mathcal{X} = \mathcal{U}, \mathcal{H} = \{y \in \mathbb{R}^d : y = \sum_{k=0}^t F^k Gu \text{ for some } u \in \mathcal{U} \text{ and } t \in \mathbb{N}\}, A(h) = Fh, B(x) = Gx$, and $f_t(h_t) = c_t(\sum_{k=1}^t F^k Gu_{t-k} + \sum_{k=1}^t F^k w_{t-k})$. Note that $\mathcal{H}, A$, and $B$ are all finite-dimensional.

**Data.** We set the time horizon $T = 750$ and dimension $d = 2$. We sample the disturbances $\{w_t\}$ from a standard normal distribution. We set the system matrix $G$ to be the identity and the system matrix $F$ to be a diagonal plus upper triangular matrix with the diagonal entries equal to $\rho$ and the upper triangular entries equal to $\alpha$. We run simulations with various values of $\rho$ and $\alpha$.

**Implementation.** We use the `cvxpy` library [Diamond and Boyd, 2016, Agrawal et al., 2018] for implementing Algorithm 1. We use step-sizes according to Theorems 3.1 and 3.3. We run the experiments on a standard laptop.

---

[2]https://github.com/raunakkmr/oco-with-memory-code.

**Results.** We compare the regret with respect to the optimal control input of OCO with unbounded memory and OCO with finite memory for various memory lengths $m$ in Fig. 3 for $\rho = 0.90$ and Fig. 4 for $\rho = 0.95$. There are a few important takeaways.

1. OCO with unbounded memory either performs as well as or better than OCO with finite memory, and it does so at comparable computational cost (Appendix G). In fact, the regret curve for OCO with unbounded memory reaches an asymptote whereas this is not the case for OCO with finite memory for a variety of memory lengths.

2. Knowledge of the spectral radius of $F$, $\rho$, is not sufficient to tune the memory length $m$ for OCO with finite memory. This is illustrated by comparing Figs. 3a to 3d. Even though small memory lengths perform well when the upper triangular value is small, they perform poorly when the upper triangular value is large. In contrast, OCO with unbounded memory performs well in all cases.

3. For a fixed memory length, OCO with unbounded memory eventually performs better than OCO with finite memory. This is illustrated by comparing Figs. 3a to 3d.

4. As we increase the memory length, the performance of OCO with finite memory eventually approaches that of OCO with unbounded memory. However, an advantage of OCO with unbounded memory is that it does not require tuning the memory length. For example, when $\rho = 0.90$ and the upper triangular entry of $F = 0.10$, OCO with finite memory with $m = 4$ performs comparably to $m = 8$ and $m = 16$ (Fig. 3c). However, when the upper triangular entry of $F = 0.12$, then it performs much worse (Fig. 3d). However, OCO with unbounded memory performs well in all cases without the need for tuning an additional hyperparameter in the form of memory length.

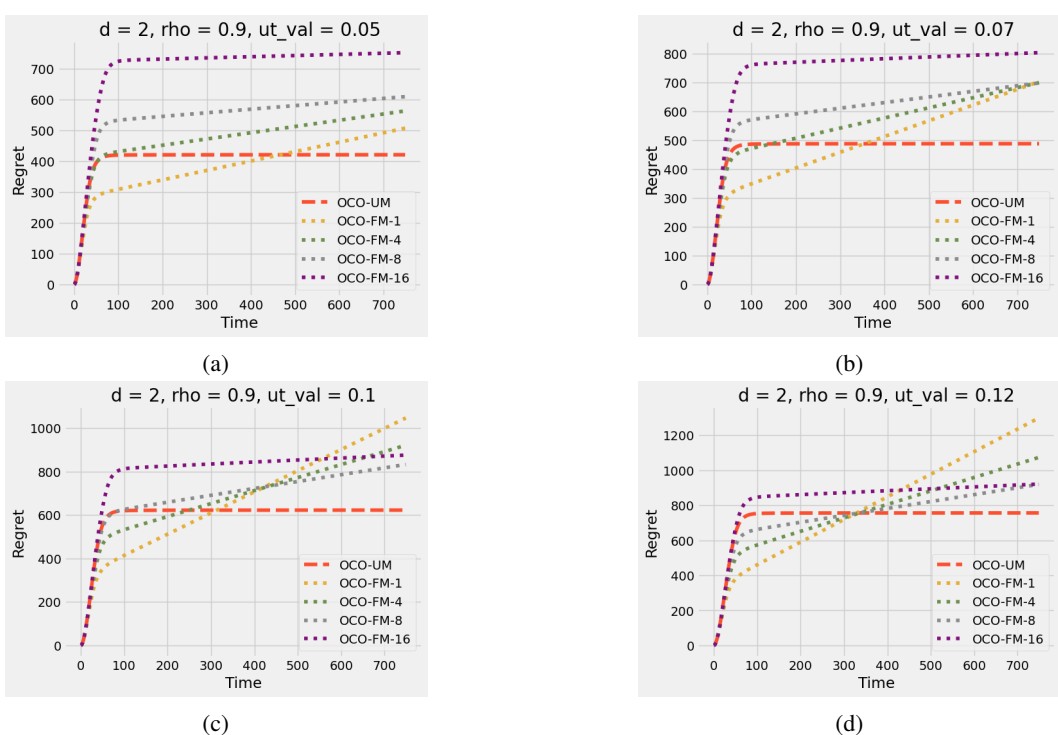

(a)  (b)  (c)  (d)

Figure 3: Regret plot for $\rho = 0.90$. The label `OCO-UM` refers to formulating the problem as an OCO with unbounded memory problem. The `OCO-FM-m` refers to formulating the problem as an OCO with finite memory problem with constant memory length $m$. The titles of the plots indicate the values of the dimension, the diagonal entries of $F$, and the upper triangular entries of $F$.

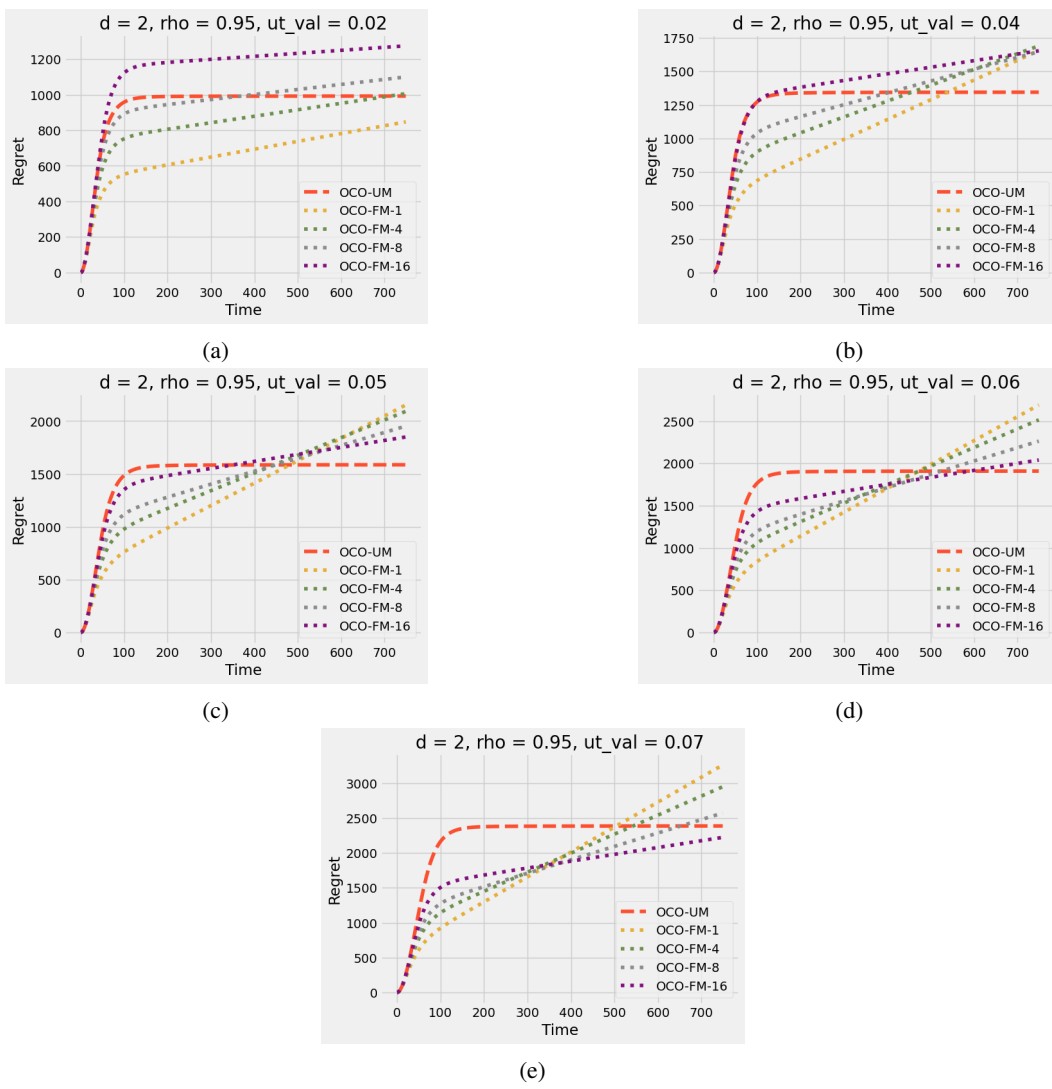

Figure 4: Regret plot for $\rho = 0.95$. The label `OCO-UM` refers to formulating the problem as an OCO with unbounded memory problem. The `OCO-FM-m` refers to formulating the problem as an OCO with finite memory problem with constant memory length $m$. The titles of the plots indicate the values of the dimension, the diagonal entries of $F$, and the upper triangular entries of $F$.

