# OpenReview forum: "Online Convex Optimization with Unbounded Memory"
_NeurIPS.cc/2023/Conference — NeurIPS 2023 poster_

### Official Review · Reviewer_58kP · 2023-06-12

**Soundness:** 3 good
**Presentation:** 3 good
**Contribution:** 2 fair
**Rating:** 3
**Confidence:** 4

**Summary:**

This paper focuses on online convex optimization with memory, a topic with increasing attention recently. Traditional framework assumes that the current environment is only affected by the decisions of a limited past, while this work considers that the current environment is affected by \emph{all} previous decisions. Specifically, the authors generalize the existing framework by studying the sequences of decisions in a typical sequence space. The authors then define the notion of memory capacity with the help of a bounded weighted norm in such sequence space and achieve new policy regret bounds. The authors verify its applicability by reducing several problems into the proposed framework, including variants of online linear control and performative prediction.

**Strengths:**

(1)	The motivation is meaningful. It is definitely important in online learning literature to capture the historical impact in the sequence.

(2)	The solution is simple, and the proof is clear and correct.

**Weaknesses:**

(1)	My first concern comes from the novelty of the proposed framework. Although it enables an infinite memory length, the impact of history is modelled as typical linear operators, and positive results are obtained only when the operator behaves like a geometric combination of past decisions, which has been studied in some areas like linear control and reinforcement learning. Could the authors provide more special cases (in sec. 2.3), or elaborate more (in the paragraph just above def 2.3), to show that the proposed framework does give some new intuition to this problem?

(2)	The technical contribution seems insufficient. The results mostly follows that of the traditional proofs in (Anava et al., 2015), with the operations on a finite sequence replaced by linear operators in functional space of sequences. Could the authors highlight the technical contributions in the proofs of their theory? For example, the explanation of (theorem 3.4) is quite clear and intuitive, is there something worth highlighted in the main results of the upper bounds?


**Questions:**

Please see the comments above.

**Limitations:**

Please see the comments above.

---

> ### Author Rebuttal · Authors · 2023-08-08
>
> Thank you for your review. We are glad you found the motivation meaningful, and the liked the simplicity and clarity of our proofs.
>
> ## Novelty of the extension from OCO with finite memory to OCO with unbounded memory and technical contributions.
>
>   * Our formulation of OCO with unbounded memory bears a strong resemblance to online linear control with adversarial disturbances. In fact, you would be correct in asserting that our setting is a special case of online linear control with adversarial disturbances **if the latter problem allowed for an infinite-dimensional state space**. However, the treatment of online linear control in [Agarwal et al., 2019b] and follow-up work is limited to finite-dimensional state spaces because their regret bounds include a polynomial dependence on a dimension-dependent constant that makes them inapplicable to problems with infinite-dimensional state unless one makes additional assumptions. One of the key contributions of our work is to find the right generalization of finite-dimensionality (namely, finite effective memory capacity) that enables generalizing these results to problems with infinite-dimensional state.
>
>   * Our OCO with unbounded memory framework and upper bound (Algorithm 1 and its analysis) might seem like simple extensions of their OCO with finite memory counterparts. However, we believe it is a feature that our framework and upper bound provide a clean abstraction for the user, while at the same time giving them a lot of power and hiding the technical details. For instance, our framework allows the user to define non-standard norms on the decision and history spaces. This can be a simple but powerful way of encoding prior knowledge about a problem. The technical complications that arise from this are captured in bounding the relevant quantities of interest, e.g., the Lipschitz constant $\tilde{L}$, the operator norm $\| A \|$, etc. Indeed, consider the application to online linear control with adversarial disturbances (section 4.1). Our seemingly simple framework and upper bound applied to this problem (Theorem 4.1 and Appendix E.3) improve upon the existing upper bound, which used a finite memory approximation. See Lemmas E.2 and E.6 for an illustration of the technical details involved when using non-standard norms, e.g., lines 885 - 891 in the proof of Lemma E.6.
>
>   * One of our main technical contributions is the **first lower bound** for OCO with finite memory, and therefore, for OCO with unbounded memory. Furthermore, this **lower bound is tight**. While the proof of the upper bound is an extension of existing proofs, our tight lower bound, which was previously unknown and uses new technical ideas, shows that the upper bound is unimprovable in the worst-case. Therefore, without additional assumptions, no other algorithm or proof technique can improve the upper bound in the worst-case. (We also provide an explicit proof of the lower bound for OCO with $\rho$-discounted infinite memory problem in Theorem D.2 in the appendix.)
>
>   * Another technical contribution is the upper bound for online linear control with adversarial disturbances (Theorem 4.1, lines 328 - 332), which improves upon existing results (lines 328 - 330 and Appendix E.3). Our regret bound (Theorem 4.1, lines 328 - 332) quantitatively improves upon the existing one (lines 328 - 330 and Appendix E.3). This is possible due to a novel use of defining weighted norms on the history and decision spaces, and using that to bound the relevant quantities in the upper bound (Lemmas E.2 and E.6).

---

> > ### Comment · Reviewer_58kP · 2023-08-12
> >
> > Thank you for the response. I understand the points that you argue, and admit that the proposed lower bound is new. But to be very honest, I am still concerning the novelty of this paper.
> >
> > On the techniques to derive the framework, it is not surprising to me to replace the assumption of "bounded regularity in each direction" in finite-dimension cases with the assumption of a bounded norm of coefficient matrices in infinite-dimension cases.
> >
> > On the novelty of the framework, I still keep my point that the current version is not enough to show its potential to give new ideas in designing algorithms. In particular, could the authors provide more special cases that the proposed framework can recover (while the existing works cannot)?

---

> > > ### Author Response · Authors · 2023-08-12
> > >
> > > Re: special cases - In addition to dynamics that exhibit geometric decay, our framework also captures examples that exhibit a transient behavior. Even though we use a scalar decay factor in our OCO with infinite memory example for simplicity, a more general example is when the decisions are multiplied by a matrix Z with spectral norm less than one. Dynamics describing a vector rotating with decreasing norm have operator norm and spectral radius less than 1. However, by stretching the space, the dynamics would be rotating according to an ellipse, so the norm would be alternatively growing and shrinking even as the vector eventually decays to the origin – as a result the transients are nontrivial and the operator norm is larger than 1 even as the spectral radius is less than 1. This is another simple example that our proposed framework can recover.

---

### Official Review · Reviewer_xqDp · 2023-06-30

**Soundness:** 3 good
**Presentation:** 2 fair
**Contribution:** 2 fair
**Rating:** 4
**Confidence:** 3

**Summary:**

This paper introduces a problem called online convex optimization (OCO) with unbound memory, which generalizes an existing problem called OCO with memory. To address this problem, the authors propose an algorithm called FTRL and analyze its regret. Moreover, the authors demonstrate that this new problem and their algorithm have several applications.

**Strengths:**

1) The problem of online convex optimization (OCO) with unbound memory is more general than the existing problem called OCO with memory.
2) The proposed algorithm enjoys a regret bound and has some applications.
3) It seems that the results in this paper can simplify the regret anlysis for the problem of online linear control.

**Weaknesses:**

1) The extension from OCO with memory to OCO with unbound memory seems to be straightforward. Moreover, the algorithm for OCO with unbound memory and the corresponding analysis are very similar to existing algorithms for OCO with memory and their analysis. So, to some extent, the novelty of this paper is limited.
2) Although OCO with unbound memory seems to be more challenging, the authors only consider the history determined by fixed linear operators, i.e., $A$ and $B$ in the paper. Moreover, the authors do not explain why they only consider this case.
3) In the experiments, the authors simply set the step size of the proposed algorithm and the existing algorithms as $1/\\sqrt{T}$, instead of tuning it according to their theoretical guarantees. In this way, it is not clear whether the improvement in experiments is consistent with their theoretical guarantees.

**Questions:**

1) Please explain the reasons for only considering the history determined by fixed linear operators.
2) It would be better if the authors could conduct experiments by setting the parameters of algorithms according to their theoretical guarantees.

**Limitations:**

There does not exist a potential negative societal impact.

---

> ### Author Rebuttal · Authors · 2023-08-08
>
> Thank you for your review. We are glad that you liked the generality of our framework and the simplification of the results for online linear control.
>
> ## Novelty of the extension from OCO with finite memory to OCO with unbounded memory.
>
>   * Our OCO with unbounded memory framework and upper bound (Algorithm 1 and its analysis) might seem like simple extensions of their OCO with finite memory counterparts. However, we believe it is a feature that our framework and upper bound provide a clean abstraction for the user that they can use for a variety of applications. For instance, our framework allows the user to define non-standard norms on the decision and history spaces. This can be a simple but powerful way of encoding prior knowledge about a problem. The technical complications that arise from this are captured in bounding the relevant quantities of interest, e.g., the Lipschitz constant $\tilde{L}$, the operator norm $\| A \|$, etc. Indeed, consider the application to online linear control with adversarial disturbances (section 4.1). Our seemingly simple framework and upper bound applied to this problem (Theorem 4.1 and Appendix E.3) improve upon the existing upper bound, which used a finite memory approximation. See Lemmas E.2 and E.6 for an illustration of the technical details involved when using non-standard norms, e.g., lines 885 - 891 in the proof of Lemma E.6.
>
>   * One of our main technical contributions is the **first lower bound** for OCO with finite memory, and therefore, for OCO with unbounded memory. Furthermore, this **lower bound is tight**. Our tight lower bound, which was previously unknown and uses new technical ideas, shows that the upper bound is unimprovable in the worst-case. Therefore, without additional assumptions, no other algorithm or proof technique can improve the upper bound in the worst-case. (We also provide an explicit proof of the lower bound for OCO with $\rho$-discounted infinite memory problem in Theorem D.2 in the appendix.)
>
>   * As alluded to above, another technical contribution is the upper bound for online linear control with adversarial disturbances (Theorem 4.1, lines 328 - 332), which improves upon existing results (lines 328 - 330 and Appendix E.3). Our regret bound (Theorem 4.1, lines 328 - 332) quantitatively improves upon the existing one (lines 328 - 330 and Appendix E.3). This is possible due to a novel use of defining weighted norms on the history and decision spaces, and using that to bound the relevant quantities in the upper bound (Lemmas E.2 and E.6).
>
> ## History is determined by fixed linear operators.
>
>   * We consider linear operators because the composition of such operators with convex functions remains convex. Nonlinear dynamics would lead to non-convexity. That is, if $f_t$ is convex, then $\tilde{f}\_t(x) = f_t ( \sum_{s=0}^{t-1} A^s B x )$ is convex if $A$ and $B$ are linear operators. Instead, if the history evolved according to nonlinear dynamics, then $\tilde{f}_t(x)$ would be $f_t$ applied to a $t$-fold composition of nonlinear operators acting on $x$, and this may not be convex.
>
> ## Choice of the step-size in the experiments.
>
>   * We have attached a PDF to the global response with after running the experiments with the theoretically optimal step-size. We will update the revision with these experiments.

---

> > ### Comment · Reviewer_xqDp · 2023-08-11
> >
> > Thanks for your response.

---

### Official Review · Reviewer_i8qJ · 2023-07-06

**Soundness:** 3 good
**Presentation:** 3 good
**Contribution:** 3 good
**Rating:** 6
**Confidence:** 3

**Summary:**

The paper considers an online learning problem between a learner and adversary. The learner chooses action $x_t$ each round, and the state $h_t$ evolves according to the dynamics $h_t = Ah_{t-1} + Bx_t$. The oblivious adversary commits to a loss function $f_t$ each round. The learner suffers cumulative loss $\sum f_t(h_t)$. This paper uses FTRL to solve this problem.

**Strengths:**

The paper studies online learning with a linear control component. The framework captures a setting where past decisions affect future loss. The paper presented theoretical results with upper and lower bounds.

The paper showed that the difficulty of this problem is captured by the so-called $p$-effective memory, which essentially quantifies the memory(less) property of the operator $A$.

**Weaknesses:**

While the problem proposed in this paper is seemingly new, it does seem to shed much new algorithmic ideas and insights.



**Questions:**

na

**Limitations:**

The authors addressed limitations and directions for future work.

---

> ### Author Rebuttal · Authors · 2023-08-08
>
> Thank you for your review. We are glad that you liked the definition of effective memory capacity, and tight upper and lower bounds on regret.
>
> ## "While the problem proposed in this paper is seemingly new, it does seem to shed much new algorithmic ideas and insights."
>
>   * From context (i.e., the fact that this sentence was listed under "Weaknesses") we suspect the reviewer meant to write "_doesn't seem to shed much new algorithmic ideas and insights_" rather than "_does_ seem to shed...". It's true that the algorithmic ideas and insights in our paper are strongly influenced by Anava et al.'s (2015) use of FTRL to solve online convex optimization with finite memory. An important innovation in our work is the use of weighted norms to prove regret bounds in the case of linear sequence dynamics, which allows us to derive non-trivial regret bounds in the case of unbounded-length histories and even leads to improved regret bounds in the case of online linear control (lines 328 - 330 and Appendix E.3). A brief summary of the novelty and technical contributions is as follows:
>
>     * Our OCO with unbounded memory framework and upper bound (Algorithm 1 and its analysis) might seem like simple extensions of their OCO with finite memory counterparts. However, we believe it is a feature that our framework and upper bound provide a clean abstraction for the user that they can use for a variety of applications. For instance, our framework allows the user to define non-standard norms on the decision and history spaces. This can be a simple but powerful way of encoding prior knowledge about a problem. The technical complications that arise from this are captured in bounding the relevant quantities of interest, e.g., the Lipschitz constant $\tilde{L}$, the operator norm $\| A \|$, etc. Indeed, consider the application to online linear control with adversarial disturbances (section 4.1). Our seemingly simple framework and upper bound applied to this problem (Theorem 4.1 and Appendix E.3) improve upon the existing upper bound, which used a finite memory approximation. See Lemmas E.2 and E.6 for an illustration of the technical details involved when using non-standard norms, e.g., lines 885 - 891 in the proof of Lemma E.6.
>
>     * One of our main technical contributions is the **first lower bound** for OCO with finite memory, and therefore, for OCO with unbounded memory. Furthermore, this **lower bound is tight**. Our tight lower bound, which was previously unknown and uses new technical ideas, shows that the upper bound is unimprovable in the worst-case. Therefore, without additional assumptions, no other algorithm or proof technique can improve the upper bound in the worst-case. (We also provide an explicit proof of the lower bound for OCO with $\rho$-discounted infinite memory problem in Theorem D.2 in the appendix.)
>
>     * As alluded to above, another technical contribution is the upper bound for online linear control with adversarial disturbances (Theorem 4.1, lines 328 - 332), which improves upon existing results (lines 328 - 330 and Appendix E.3). Our regret bound (Theorem 4.1, lines 328 - 332) quantitatively improves upon the existing one (lines 328 - 330 and Appendix E.3). This is possible due to a novel use of defining weighted norms on the history and decision spaces, and using that to bound the relevant quantities in the upper bound (Lemmas E.2 and E.6).

---

> > ### Comment · Reviewer_i8qJ · 2023-08-18
> >
> > Thanks for the response ( yes I did mean to say 'doesn't', apologies for the typo ). I think this is a technically sound paper and will keep my original score. I will also read other reviewers' comments.

---

### Official Review · Reviewer_cA1Y · 2023-07-08

**Soundness:** 3 good
**Presentation:** 3 good
**Contribution:** 2 fair
**Rating:** 6
**Confidence:** 3

**Summary:**

This paper studies a generalization of online convex optimization (OCO) with memory. The setting allows the current stage cost to depend on all past decisions via a discrete-time linear dynamical system. The authors proposed a follow-the-regularized-leader algorithm that can achieve a sublinear static regret against any fixed action. They also showed a lower bound that matches the regret upper bound in the order of horizon $T$ and Lipschitz constants. The authors discussed two applications of their results to online control and online performative prediction.

**Strengths:**

Theory for online (convex) optimization with unbounded memory is important in the field of learning for control. While this problem is intractable in general, it is good to see results that formally define the “effective memory” and study the corresponding upper/lower bounds.

**Weaknesses:**

My major concern about this work is about the problem setting: I believe the setting proposed here is a special case of online control with adversarial disturbances (e.g., [Agarwal et al., 2019b]). Specifically, the history $h_t$ corresponds to the state and the decision $x_t$ corresponds to the control input. The only difference might be the benchmark: While online control compares against the best DAC policy, this work compares with a fixed action. But the DAC policy class can easily contain any fixed action if we add a dummy dimension with entry 1 in the (adversarial) disturbances. I hope the authors can correct me if my understanding is wrong, and a discussion in the revision may be helpful.

Since the proposed problem setting can be reduced to (or maybe equivalent to) online control with adversarial disturbances, one should evaluate the results in this work by comparing with not only [Agarwal et al., 2019b], but also more recent works on online control like [Minasyan et al., 2022], [Chen et al., 2022], and [Lin et al., 2022]. To the best of my knowledge, existing results can handle much more complicated settings that involves time-varying dynamics, unknown dynamics, or even nonlinear dynamics. They considered stronger benchmarks like adaptive regret or dynamic regret. I encourage the authors to do a more detailed literature review about online control and clarify the significance of the main results.

Besides, I also have a concern about the time complexity of the proposed algorithm. It seems that the time/memory complexity of constructing $\tilde{f}_t$ from past $f_1, \cdots, f_t$ grows linearly with respect to time $t$. However, existing online control algorithms like the one in [Agarwal et al., 2019b] only requires $log(T)$ time/memory for each decision. Thus, I am not sure how practical the proposed FTRL algorithm is.

[Minasyan et al., 2022]: https://arxiv.org/pdf/2202.07890.pdf
[Chen et al., 2022]: https://arxiv.org/pdf/2110.07807.pdf
[Lin et al., 2022]: https://arxiv.org/pdf/2210.12320v1.pdf


**Questions:**

Please see my comments in the previous section. I gave the score based on my current understanding about the similarity with online control, and I would be happy to raise the score if the authors address my concern.

**Limitations:**

The authors discussed about some future directions in Section 5.

---

> ### Author Rebuttal · Authors · 2023-08-08
>
> Thank you for your review. We are glad that you liked the definition of effective memory capacity, and tight upper and lower bounds on regret.
>
> ## Application to Online Linear Control with Adversarial Disturbances.
>   * You are correct that our formulation of OCO with unbounded memory bears a strong resemblance to online linear control with adversarial disturbances (OLC) with a fixed control input. In fact, you would be correct in asserting that our setting is a special case of OLC **if the latter problem allowed for an infinite-dimensional state space**. However, the treatment of OLC in [Agarwal et al., 2019b] and follow-up work is limited to finite-dimensional state spaces because their regret bounds include a polynomial dependence on a dimension-dependent constant that makes them inapplicable to problems with infinite-dimensional states unless one makes additional assumptions. One of the key contributions of our work is to find the right generalization of finite-dimensionality (namely, finite effective memory capacity) that enables generalizing these results to problems with infinite-dimensional states.
>   * Our framework is *not* a special case of OLC. In fact, in lines 305-330 we show the reverse.
>     * Decisions in our framework correspond to policies in the linear control framework, not a fixed control input. The decision is a DAC policy (Definition 4.1) defined by a sequence of matrices and a fixed matrix (line 294).
>     * Our DAC policy acts on the entire history of past disturbances, whereas the "truncated" DAC policy used in [Agarwal et al., 2019b] only acts on a fixed, constant number of past disturbances. The class of strongly-stable linear controllers is a subset of our DAC policy class, but not a subset of the "truncated" DAC policy class. (See [Agarwal et al., 2019a, Section 16.5].) Unbounded-length DAC policies are why modelling an infinite-dimensional space is relevant.
>     * Our regret bound (Theorem 4.1, lines 328 - 332) improves upon the existing one by $O(d (\log T)^{3.5} \kappa^5 (1-\rho)^{-1})$ (Appendix E.3). This is possible due to a novel use of defining weighted norms on the history and decision spaces, and using that to bound the relevant quantities in the upper bound (Lemmas E.2 and E.6).
>     * A "truncated" DAC policy is a sequence of $d \times d$ matrices of length $2 \kappa^4 (1-\rho)^{-1} \log T$ [Agarwal et al., 2019b]. Our DAC policy is a sequence of $d \times d$ matrices of unbounded length. Yet, we capture the dimension of this infinite-dimensional space in a way that still improves the overall bound, including completely eliminating the dependence on $\log T$, and improving the dependence on $d, \kappa$ and $(1-\rho)$ (Theorem 4.1 and Appendix E.3).
>   * Thank you for citing the additional works. We will add them to the revision. However, all of them analyze the problem under a finite memory approximation even though it is inherently an unbounded memory problem. We focused on addressing gaps in the OCO with memory literature by first developing the general framework of OCO with unbounded memory. Then, we proved an upper bound and a **tight lower bound** on the regret, including a previously unknown, tight lower bound for OCO with finite memory. The basic online control setting is just one application - we also consider an application to a performative prediction problem, showing how our general framework can unify two seemingly disparate areas of work. The value of our work lies in (i) an improvement in the upper bound for control (Theorem 4.1); (ii) a simplification of the regret analysis; (iii) a new lens to study extensions that you cited. Our improvements will carry over to these extensions after developing appropriate extensions of our framework, and it is an important direction for future work.
>
> ## Time complexity of the proposed algorithm.
>   * We provide details for efficient implementation of Algorithm 1 in Appendix G. Here, we summarize how it only has a constant overhead compared to standard FTRL and FTRL for OCO with finite memory (Algorithm 1 in Anava et al.)
>     * Our Algorithm 1 chooses iterate $x_{t+1}$ as the minimizer of $\sum_{s=1}^t f_s(\sum_{k=0}^{s-1} A^k B x) + \frac{R(x)}{\eta}$. Each such minimization problem requires evaluating $\sum_{s=1}^t f_s(\sum_{k=0}^{s-1} A^k B x)$. In Appendix G, we show how to do this in $O(t)$ time by iteratively updating the function arguments. Here, $O(\cdot)$ notation hides constant factors excluding $t$ and $T$ but includes constant factors related to the dimensionality of the decision space. See Appendix G for details.
>     * Standard FTRL and Algorithm 1 of Anava et al. also choose iterate $x_{t+1}$ by minimizing over a sum of $t$ functions. For example, Algorithm 1 of Anava et al. chooses iterate $x_{t+1}$ as the minimizer of $\sum_{s=1}^t f_s(x, \dots, x) + \frac{R(x)}{\eta}$. Each such minimization problem requires evaluating $\sum_{s=1}^t f_s(x, \dots, x)$. This takes $O(t)$ time, where the $O(\cdot)$ notation hides constant factors excluding $t$ and $T$ but includes constant factors related to the dimensionality of the decision space.
>   * For OLC specifically, the squared-norm regularizer $R$ (Lemma E.4) results in an update rule similar in spirit to online gradient descent and the existing algorithm in [Agarwal et al., 2019b]. The dimensionality of our decision space depends on $O(T)$, whereas it depends on $O(\log T)$ in [Agarwal et al., 2019b]. This is the only factor in $T$ that leads to a difference between the runtimes of the two algorithms. One can change our decision space to be the "truncated" DAC policy class and the runtime will depend on $O(\log T)$. This will introduce an error term that is the difference between the costs of the best strongly-stable linear controller and of the best "truncated" DAC policy. We can bound this using Lemma 5.2 of [Agarwal et al., 2019b]. This is not the dominant term in the regret bound (Appendix E.3 of our paper), so the final bound is unchanged.

---

> > ### Comment · Reviewer_cA1Y · 2023-08-12
> > **Thank you for the detailed response!**
> >
> > I want to thank the authors for the detailed response. I apologize for ignoring the difference in the infinite/finite dimension issue in my initial review. I believe allowing the state (or history) space to have infinite dimensions is a good technical contribution, and I would appreciate it if the authors can elaborate more about the significance of this contribution.
> >
> > 1. I’m not fully convinced about the necessity of allowing the state (or history) space to have infinite dimensions. From the current two examples, it seems like the infinite-dimension state (or history) space is a result of using a specific proof technique, while other techniques may solve the same problems in finite dimensions.
> >
> > 2. If I understand correctly, one can still apply the algorithm in [Agarwal et al., 2019b] to the setting of this paper, but the regret will be unbounded when the state (or history) space has infinite dimensions. Is the dependence on the dimension a fundamental limit of the algorithm in [Agarwal et al., 2019b] or due to the proof technique?
> >
> > 3. The computational cost of the algorithm depends on how fast one can implement an oracle call to the operator A. I’m not sure if the oracle $O_A$ can always run in a constant time especially when the dimension of the state (or history) space is infinite.

---

> > > ### Author Response · Authors · 2023-08-14
> > >
> > > 1.
> > > * Infinite dimensional objects arise from finite dimensional control problems when we use a **policy** to react to disturbances.
> > >   * The linear control problem is not convex unless we lift from system state space into the (infinite dimensional) DAC policy space. This is *not* a result of a proof technique and has algorithmic implications.
> > >   * The linear dynamics that define the control problem have a finite ($d$) dimensional representation. However, the presence of external disturbances means that we need to model the *system response* rather than merely the *system state*. The system response is infinite dimensional. For example, see [System Level Synthesis, Anderson et al., 2019](https://arxiv.org/abs/1904.01634).
> > >     * To compute how policies (decisions) affect costs (losses) through the function $\tilde{f}_t$, we need to model the system response. So, this is an algorithmic consideration and not just the result of a proof technique.
> > >     * [Agarwal et al., 2019b] consider a *truncated* system response of length $O(\log T)$ that is an approximation. This is also where they appeal to results for OCO with finite memory, which our paper should be seen as a replacement for.
> > >   * More generally, other examples of infinite dimensional history/states include infinite impulse response filters in signal processing (e.g., elliptic filters).
> > > * To contrast, suppose instead of choosing policies, we were interested in a "fixed constant input" model of control. That is, the learner chooses a fixed control input in each round and the benchmark is the best fixed control input. This would indeed result in a finite dimensional history space.
> > >   * The history combines "noiseless" state with action: $\bar{s}_{t+1} F \bar{s}_t + G a_t, h_t = (\bar{s}_t, a_t)$.
> > >   * The linear operators are block matrices: $A = [F, 0; 0, 0]$ and $B = [G; I] / \| [G; I] \|$. (The scaling on $B$ is to ensure that it has unit norm, which we assume in our paper for convenience.)
> > >   * The loss functions are $f_t(h_t) = c_t( \bar{s}\_t + \sum_{k=1}^t F^k G w_{t-k} , a_t) = c_t(s_t, a_t)$.
> > >
> > > 2. Yes, you are correct. The algorithm of [Agarwal et al., 2019b] and their proofs work for standard Euclidean norms and the corresponding matrix norms. If the state and history spaces are infinite dimensional with non-standard norms, then there is a mismatch between the problem setup and their algorithm and proofs. So, in some sense the regret is unbounded because of both the algorithm and the proof technique. The algorithm would use a regularizer that is strongly convex with respect to the Euclidean norm, resulting in suboptimal regret compared to our algorithm that uses a regularizer adapted to the appropriate norm; the proofs would compute the Euclidean norm of infinite dimensional objects.
> > >
> > > 3. We agree with your point.
> > >     * See lines 981 - 988 in the appendix where we discuss this issue. We believe that it is a feature that our framework provides a clean abstraction for the user that they can use for a variety of applications. The dimensionality of $\mathcal{X}$, the choice of the operator $A$, etc. are application dependent. The user could use our framework with a lower dimensional decision space $\mathcal{X}'$ and then analyze the error that results from such an approximation. Meanwhile, our framework allows the user to define non-standard norms on the decision and history spaces. This can be a simple but powerful way of encoding prior knowledge about the application, and our framework handles the technical complications that arise from this (e.g., bounding the Lipschitz constant, operator norms, etc. - see Lemmas E.2 and E.6 (especially lines 885 - 891) for an illustration).

---

> > > > ### Comment · Reviewer_cA1Y · 2023-08-14
> > > >
> > > > Thank you for responding to my follow-up questions. I have not seen much previous work on online learning/control that emphasizes an infinite-dimensional state space, so I encourage the authors to dig deeper if this can be another spotlight of the paper. I will raise my score because the major concern I mentioned in my initial review has been addressed.

---

### Author Rebuttal · Authors · 2023-08-08

We have responded to each reviewer individually. This global rebuttal only includes plots for the experiments requested by Reviewer xqDp.

---

### Decision · Program_Chairs · 2023-09-21

**Decision:**

Accept (poster)

**Comment:**

This paper attacks OCO with memory by casting the history into a banach space and applying online learning in that space. A key application is in adversarial control, where considering the appropriate Banach space allows the authors to reduce a dimension factor in previous bounds. Theory-oriented papers can have significant value along several axes, including (but not limited to): (1) generating novel technical approaches, (2) improving results on known and well-studied problems,  and (3) developing a novel and interesting setting of study. Some reviewers raise concerns about (1) and (3): the techniques used in the paper are relatively standard, and the setting of adversarial control is also well-studied. However, this  paper satisfies (2) by improving the general OCO with memory bounds and applying them to improve the adversarial control bounds by a polynomial factor in the dimension. Such dimension factor improvements are relatively rare and so this represents a significant enough contribution to be accepted.